# Persistence, period and precision of autonomous cellular oscillators from the zebrafish segmentation clock

Alexis B Webb[1,2†], Iván M Lengyel[3], David J Jörg[4], Guillaume Valentin[1,2‡], Frank Jülicher[4], Luis G Morelli[3§], Andrew C Oates[1,2,5*†]

[1]MRC-National Institute for Medical Research, London, United Kingdom; [2]Max Planck Institute of Molecular Cell Biology and Genetics, Dresden, Germany; [3]Departamento de Física, FCEyN UBA and IFIBA, CONICET, Buenos Aires, Argentina; [4]Max Planck Institute for the Physics of Complex Systems, Dresden, Germany; [5]Department of Cell and Developmental Biology, University College London, London, United Kingdom

**Abstract** In vertebrate development, the sequential and rhythmic segmentation of the body axis is regulated by a "segmentation clock". This clock is comprised of a population of coordinated oscillating cells that together produce rhythmic gene expression patterns in the embryo. Whether individual cells autonomously maintain oscillations, or whether oscillations depend on signals from neighboring cells is unknown. Using a transgenic zebrafish reporter line for the cyclic transcription factor Her1, we recorded single tailbud cells in vitro. We demonstrate that individual cells can behave as autonomous cellular oscillators. We described the observed variability in cell behavior using a theory of generic oscillators with correlated noise. Single cells have longer periods and lower precision than the tissue, highlighting the role of collective processes in the segmentation clock. Our work reveals a population of cells from the zebrafish segmentation clock that behave as self-sustained, autonomous oscillators with distinctive noisy dynamics.

*For correspondence: andrew.oates@crick.ac.uk

Present address: [†]The Francis Crick Institute, Mill Hill Laboratory, London, United Kingdom; [‡]Genoway, Lyon, France; [§]Instituto de Investigación en Biomedicina de Buenos Aires (IBioBA) – CONICET – Partner Institute of the Max Planck Society, Buenos Aires, Argentina

## Introduction

Populations of coordinated oscillators occur in a variety of biological systems. Examples include the rhythmic flashing of fireflies, the spiral aggregation of microbes, and the daily oscillation of circadian clocks in nearly all organisms. Communication between the individual oscillators can influence whether oscillations are maintained, i.e. their persistence, as well as their period and their precision. Without examining the properties of an individual in isolation from its neighbors, a state that we define as autonomous, it is challenging to assign the relative contribution of individual and collective processes to the observed rhythmic behavior of the population.

During vertebrate embryogenesis, coordinated genetic oscillations occur in a population of cells in the posterior-most tissues of the body axis, the tailbud and presomitic mesoderm (PSM). These oscillations generate a rhythmic spatial pattern. This "segmentation clock" is thought to subdivide the embryonic body into morphological segments, called somites, which arise rhythmically and sequentially from the PSM. Persistent oscillating gene expression within the tailbud and PSM corresponds to segment formation in chick, mouse, and zebrafish (*Palmeirim et al., 1997*; *Dequéant et al., 2006*; *Krol et al., 2011*). Looking across biological systems, persistent and coherent rhythms in a population can be the product of synchronized cell-autonomous oscillators, or alternatively can be the outcome of population-level coupling of otherwise non-oscillatory cells. The autonomy of circadian clock neurons was demonstrated by recording daily oscillations in firing rate

**eLife digest** The timing and pattern of gene activity in cells can be very important. For example, precise gene activity patterns in 24-hour circadian clocks help to set daily cycles of rest and activity in organisms. In such scenarios, cells often communicate with each other to coordinate the activity of their genes. To fully understand how the behavior of the population emerges, scientists must first understand the gene activity patterns in individual cells.

Rhythmic gene activity is essential for the spinal column to form in fish and other vertebrate embryos. A group of cells that switch genes on/off in a coordinated pattern act like a clock to regulate the timing of the various steps in the process of backbone formation. However, it is not clear if each cell is able to maintain a rhythm of gene expression on their own, or whether they rely on messages from neighboring cells to achieve it.

Now, Webb et al. use time-lapse videos of individual cells isolated from the tail of zebrafish embryos to show that each cell can maintain a pattern of rhythmic activity in a gene called *Her1*. In the experiments, individual cells were removed from zebrafish and placed under a microscope to record and track the activity of *Her1* over time using fluorescent proteins. These experiments show that each cell is able to maintain a rhythmic pattern of *Her1* expression on its own.

Webb et al. then compared the *Her1* activity patterns in individual cells with the *Her1* patterns present in a larger piece of zebrafish tissue. The experiments showed that the rhythms in the individual cells are slower and less precise in their timing than in the tissue. This suggests that groups of cells must work together to create the synchronized rhythms of gene expression with the right precision and timing needed for the spinal column to be patterned correctly.

In the future, further experiment with these cells will allow researchers to investigate the genetic basis of the rhythms in single cells, and find out how individual cells work together with their neighbors to allow tissues to work properly.

and gene expression from single cells for several cycles in the absence of their neighbors (*Welsh et al., 1995*; *Webb et al., 2009*). In contrast, some microbial systems have been shown to produce oscillations only when at critical densities that allowed cell-to-cell communication, otherwise the isolated cells were not rhythmic (*Gregor et al., 2010*; *Danino et al., 2010*). Therefore, to test for autonomy of cellular oscillators in the segmentation clock, it is imperative to determine whether individual cells can oscillate in the absence of signals from their neighbors.

Historically, the term autonomy has appeared many times in the segmentation clock literature, starting with the observation that gene expression in explanted PSM can oscillate in the absence of neighboring tissues (*Palmeirim et al., 1997*; *1998*; *Maroto et al., 2005*). This means the PSM is autonomous at the tissue level. The question of whether individual segmentation clock cells are able to oscillate autonomously, that is, when fully separated from the tissue, has been debated for decades. Early theoretical arguments explored this possibility (*Cooke and Zeeman, 1976*), as well as scenarios where coupling led to oscillations (*Meinhardt, 1986*). The possibility for an auto-regulatory negative feedback loop arising from the transcription and translation of members of the Hes/Her gene family would be consistent with a cell-autonomous mechanism (*Hirata et al., 2002*; *Lewis, 2003*; *Monk, 2003*), and oscillations in this gene family have been observed across vertebrate species (*Krol et al., 2011*). However, the discovery of oscillations in the Delta-Notch system in all vertebrates and in many genes of the Wnt and FGF intercellular signaling pathways in mouse and chick, raises the possibility that communication between cells may play a critical role in the generation and/or maintenance of the oscillations (*Dequéant et al., 2006*; *Krol et al., 2011*; *Goldbeter and Pourquié, 2008*).

Two pioneering studies have attempted to address cellular autonomy in the chick and mouse segmentation clocks. In the first study, cells isolated from chick PSM, then cultured in suspension and fixed at subsequent time intervals, showed changes in cyclic gene expression (*Maroto et al., 2005*). Due to the unavoidable uncertainty in reconstructing a time series from static snapshots of different cells, the authors of this study were not able to distinguish between noisy autonomous oscillators and stochastic patterns of gene expression, and highlighted the need for real-time reporters to

investigate the autonomy of PSM cells. In the second study, the first real-time reporter of the segmentation clock, a luciferase reporter of Hes1 expression in mouse, allowed individual mouse PSM cells to be observed in vitro (*Masamizu et al., 2006*). Three cells were reported, showing at most 4 expression pulses with variable duration and amplitude, which appeared to damp out. This study concluded that PSM cells may be "unstable" oscillators, and highlighted the role of intercellular coupling for maintenance of oscillations. Reflecting this, the authors modeled the cells as excitable systems dependent on noise or signals from their neighbors for pulse generation. Thus, the degree of autonomy of completely isolated cells from the segmentation clock in mouse and chick remains unclear.

Working from the segmentation phenotypes of mutant zebrafish and the identity of the mutated genes, cases were originally made both for and against cell-autonomous oscillators in the zebrafish segmentation clock (*Jiang et al., 2000*; *Holley et al., 2002*). More recent results bring support to the idea of autonomous oscillators that are synchronized to each other. Treatment of zebrafish embryos with a γ-secretase inhibitor targeting the Notch intercellular domain (DAPT) leads to a loss of spatial coherence in oscillating gene expression (*Riedel-Kruse et al., 2007*). Additionally, imaging of individual cells in mutant embryos with reduced Notch signaling show that oscillations persist under these conditions (*Delaune et al., 2012*), though this does not rule out the possibility of Notch or other signaling factors playing a role in promoting oscillations. We recently developed an in vitro primary culture system to image gene expression in individual tailbud and PSM cells (*Webb et al., 2014*), allowing the possibility of autonomous oscillations to be tested directly.

In this paper we measure the intrinsic properties of single zebrafish cells isolated from the segmentation clock in the tailbud and show that they behave as autonomous genetic oscillators in vitro, in the absence of cell-cell or tissue-level coupling. We observe a striking variability in the cells' dynamics and find that a long-timescale noise in a theoretical description of the autonomous oscillator can account for this.

We then ask how the behavior of these cell-autonomous oscillators compares to oscillations at the level of the intact embryo. It is thought that collective processes at cellular and tissue level influence the period of segmentation in zebrafish. Theoretical analysis of the collective behavior of many cellular oscillators with time-delayed coupling shows that this process can set a collective period in a synchronized population (*Herrgen et al., 2010*). This predicts that cells isolated from the tailbud will have a different period than the population. In addition, the precision of oscillation in a synchronized population can be higher than that of component oscillators (*Herzog et al., 2004*; *Garcia-Ojalvo et al., 2004*), and this scenario has been proposed for the segmentation clock (*Masamizu et al., 2006*). However, an alternative case of synchronized oscillators with the same individual precision as the population has also been considered for the segmentation clock (*Horikawa et al., 2006*). Knowledge of the period and precision of cells isolated from the segmentation clock should enable tests of these ideas.

Using our autonomous oscillator data we characterize the period and precision of individual cells; we find that individual cells have a longer period and are less precise than the population in the tissue. Together, these results have implications for the pace-making circuit and the collective organization of the segmentation clock.

## Results

### Oscillations in isolated segmentation clock cells in vitro

To test whether single cells behave as autonomous oscillators requires a cyclic gene expression reporter and a primary cell culture system. The cyclic bHLH transcription factor Her1 has been proposed to act within a core negative feedback loop that drives oscillations (*Holley et al., 2000*; *Oates and Ho, 2002*; *Schröter et al., 2012*) and has previously been used to follow segmentation clock dynamics in the embryo (*Delaune et al., 2012*; *Soroldoni et al., 2014*). We used our transgenic zebrafish line *Looping*, which expresses a Her1-VenusYFP (Her1-YFP) fusion reporter driven from the regulatory elements of the *her1* locus with accurate temporal and spatial dynamics in the embryo (*Soroldoni et al., 2014*).

Although cells are thought to slow their oscillations as they leave the tailbud and differentiate in the PSM, progenitor cells in the tailbud are thought to maintain a regular rhythm throughout

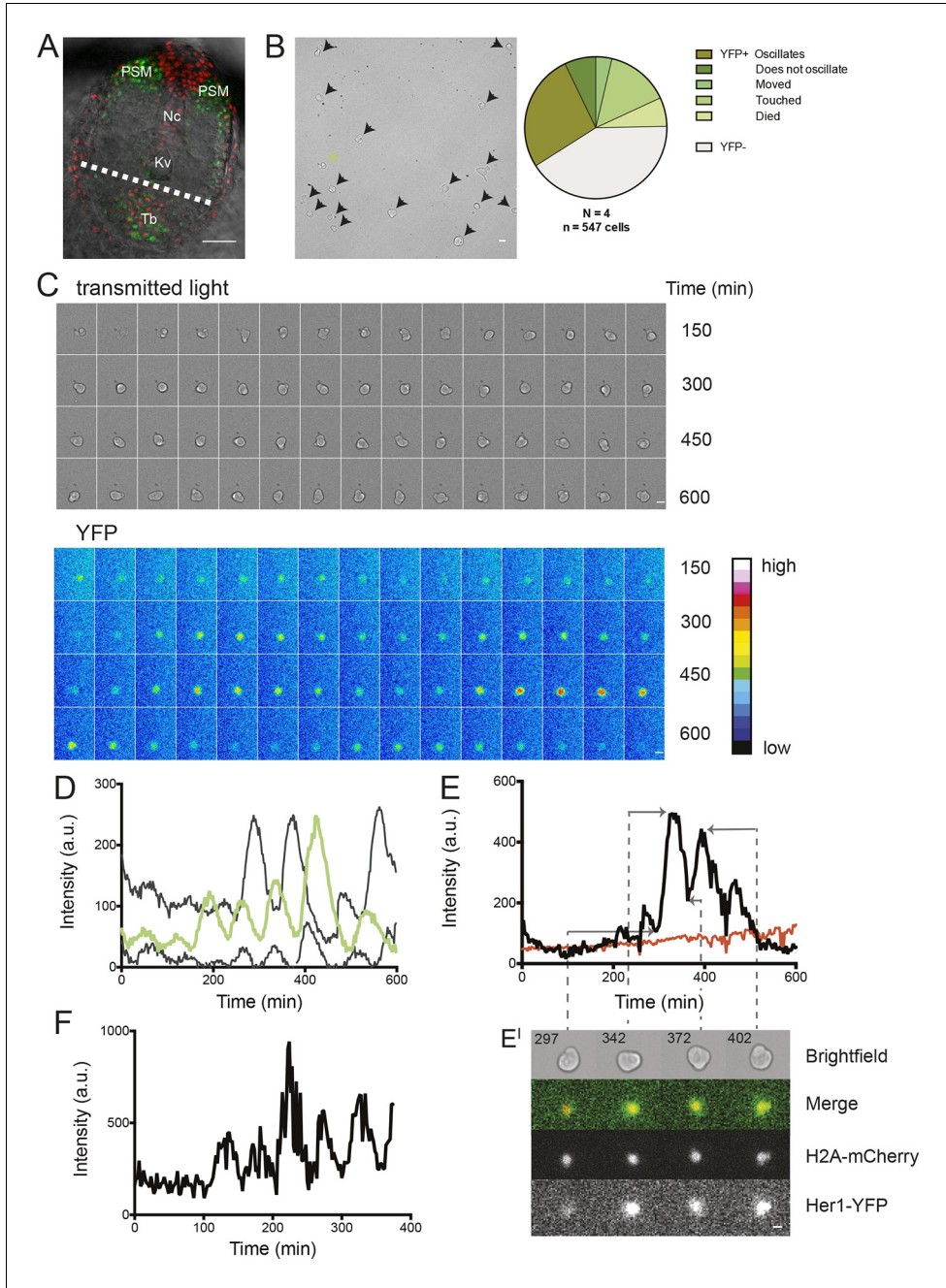

**Figure 1.** Zebrafish segmentation clock cells oscillate autonomously in culture. (**A**) Confocal section through the tailbud of a *Looping* zebrafish embryo in dorsal view where the dotted line indicates the anterior limit of tissue removed. Nuclei are shown in red and YFP expression in green. Scale bar = 50 µm. Kupffer's vesicle (Kv), notochord (Nc), presomitic mesoderm (PSM), tailbud (Tb). (**B**) A representative 40x transmitted light field with dispersed low-density *Looping* tailbud-derived cells. Individual cells highlighted with black arrowheads; green arrowhead shows cell with green time series in (**D**). Scale bar = 10 µm. Pie chart: More than half of the in vitro population of *Looping* tailbud cells (*n* = 321 out of 547 cells combined from 4 independent culture replicates as described in Materials and methods) expresses the Her1-YFP reporter. Some expressing cells are disqualified because they move out of the field of view (4%), touch other cells by colliding in the field of view (12%) or following division (2%) for a total of 14%, or do not survive until the end of the 10-hr recording (7%). (**C**) Montage of timelapse images (transmitted light, top; YFP, bottom) of a single tailbud cell (green arrowhead in panel B) over 10-hr recording. Scale bar = 10 µm. (**D**) YFP signal intensity (arbitrary units) measured by tracking a regions of interest over 3 single tailbud cells (green trace follows cell marked by green arrowhead in B, gray traces are two additional cells from culture). Plotted in 2-min intervals. (**E**) Plot of Her1-YFP (black) and H2A-mCherry (red) signal

*Figure 1 continued on next page*

*Figure 1 continued*

intensity over time measured together from a representative cell. (E'). Nuclear YFP signal accumulates and degrades over time, as shown in the overlay of H2A-mcherry signal (red channel) and Her1-YFP signal (green channel) during troughs (297, 372) and peaks (342, 402) in Her1 expression. mCherry signal in the nucleus is relatively constant. Plotted in 2-min intervals. (**F**) Plot of YFP intensity (a.u.) over time in a fully isolated tailbud cell within a single well of a 96-well plate. Plotted in 2-min intervals.

The following source data and figure supplements are available for figure 1:

**Source data 1.** Summary table of segmentation clock tissue and cellular oscillatory properties.
**Source data 2.** Summary table of low-density segmentation clock cell experiments.
**Source data 3.** Time series data from low-density segmentation clock cells.
**Figure supplement 1.** Her1-YFP-expressing cells in the zebrafish tailbud.
**Figure supplement 2.** Peak finding in time series to estimate period and amplitude.
**Figure supplement 3.** Persistent oscillations in explanted tailbud.
**Figure supplement 4.** Time series of low-density segmentation clock cells in serum-only culture.
**Figure supplement 5.** Time series of low-density segmentation clock cells.
**Figure supplement 6.** Characterization of Ntla and Tbx16 antibodies.
**Figure supplement 7.** Expression of tailbud markers in vivo and in low-density cultures of segmentation clock cells.
**Figure supplement 8.** Analysis of low-density segmentation clock cell cultures.
**Figure supplement 9.** Time series of fully isolated segmentation clock cells.

development (*Delaune et al., 2012*; *Giudicelli et al., 2007*; *Morelli et al., 2009*; *Uriu et al., 2009*; *Ay et al., 2014*; *Shih et al., 2015*). In search of these cells, we first explanted intact tailbuds from 8-somite stage embryos homozygous for the *Looping* transgene (n=3) (*Figure 1A*; *Figure 1—figure supplement 1*) into culture and recorded the Her1-YFP signal within a central region of the tissue. After local background subtraction, we generated time series of average intensities of the regions of interest, estimated period from inter-peak intervals and measured amplitudes for each cycle (*Figure 1—figure supplement 2*). We observed persistent Her1-YFP oscillations (*Figure 1—figure supplement 3*) that did not slow down (period 42.5 ± 11.4 min (mean ± SD) at 26°C) (*Figure 1—source data 1*).

We next used our primary culture protocol (*Webb et al., 2014*) to generate low-density cultures of tailbud cells from multiple *Looping* tailbuds, which were recorded for 10 hr. In identical culture conditions to the explanted tailbuds, most individual cells that expressed Her1-YFP displayed a few pulses that appeared to damp out early in the recording (*N* = 2 independent cultures, *n* = 52 total cells, median cycles = 2) (*Figure 1—figure supplement 4*; *Figure 1—source data 1*). Fgf is expressed at elevated levels in the tailbud and is proposed to play roles in maintaining an oscillatory progenitor state in the embryo (*Dubrulle et al., 2001*; *Sawada et al., 2001*; *Ishimatsu et al., 2010*; *Niwa et al., 2007*). In contrast to the serum-only treatment, when Fgf8b was added to part of the same cell suspension within a divided imaging dish, a persistent rhythmic behavior was observed (*Figure 1—figure supplement 5*). The number of cycles increased dramatically, spanning the recording interval and without altering the period (*N* = 4 independent cultures, *n* = 547 total cells; median cycles = 5; *Figure 1—source data 1*).

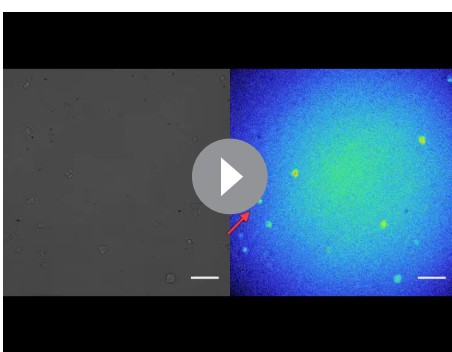

**Video 1.** Low-density segmentation clock cells oscillate in vitro. Field of view containing cell in *Figure 1B–C*, highlighted by the red arrow. This field contains 18 cells in total, with 9 expressing YFP. We observed 5 cell divisions, the highest number in any experiment, including one non-YFP cell, 3 YFP-positive cells, which are excluded from analysis because of division, and one YFP-positive cell disqualified due to contact with another cell in the field prior to the division. The remaining 5 YFP-positive cells, including the highlighted cell, are part of the low-density data set. Total duration = 10 hr; Time interval = 2 min; field size = 410 x 410 μm; Scale bar = 50 μm.

An imaged field contained 15 to 20 cells (*Figure 1B*, black arrowheads) and more than 50% of these cells (321/547) had YFP signal (*Figure 1B*, *Figure 1—source data 2*). Typically, cells remained rounded and displayed blebs, as expected from early zebrafish progenitors in vivo and in vitro (*Diz-Muñoz et al., 2010*; *Maître et al., 2012*; *Ruprecht et al., 2015*). Cells from the same embryos cultured in parallel maintained tailbud marker expression (*Figure 1—figure supplement 6*, *Figure 1—figure supplement 7*), suggesting that under these conditions the cells retain a tailbud progenitor phenotype (*Martin and Kimelman, 2010*; *2012*). For the remainder of the manuscript we focus on the Fgf8b-treated cells as a model to understand the properties of segmentation clock oscillators.

From the 321 cells that were YFP-positive at the beginning of the recording, we first excluded from analysis any cell that moved out of the imaging field, died, or touched another cell either through movement or division at any point in the recording. From the remaining 189 autonomous cells, we observed rhythmic expression (two or more pulses of expression) in 147, which we term the low-density data set (*Figure 1B–D*; *Figure 1—figure supplement 5*; *Figure 1—source data 2*; *Figure 1—source data 3*). An illustrative imaged field is shown in *Video 1*. Although some cells continued to oscillate after division, analysis of these rare events was complicated by the tendency of the daughters to strongly adhere to each other, and is beyond the scope of this study. We found a distribution of periods (78.8 ± 15.3 min [mean ± SD from 442 cycles]) and amplitudes (*Figure 1—figure supplement 8A,B*). Importantly, we observed no systematic slowing in successive oscillations in our data suggesting that the cultures were stationary over this interval (*Figure 1—figure supplement 8E*).

To rule out the possibility that oscillations in the YFP signal were influenced by focal drift, we recorded from individual cells co-expressing Her1-YFP and the nuclear marker H2A-mCherry. We found that the Her1-YFP signal co-localized with the nuclear mCherry signal, which did not oscillate (*n* = 8 cells, *Figure 1E, E'*) indicating that focal drift does not contribute strongly to changes in YFP intensity. Combined, these results show that cells from the zebrafish tailbud do not need cell-cell contact to maintain oscillations.

Nevertheless, diffusible factors could be released rhythmically in these cultures and maintain oscillations. To test the ability of a fully isolated cell to oscillate, we used serial dilution to obtain and image single tailbud cells isolated in individual culture chambers. We found that these fully isolated cells can also sustain oscillations (*N* = 5, *n* = 10, period = 62 ± 21 min (mean ± SD), median cycles = 5) (*Figure 1F*; *Figure 1—figure supplement 9*; *Figure 1—source data 1*; *Video 2*).

Together, these data reveal the existence of a population of autonomously oscillating cells from the zebrafish segmentation clock. Mechanisms of oscillation in zebrafish based on reaction-diffusion processes (*Meinhardt and Gierer, 2000*), and which rely on diffusion across the tissue and do not contain cell-autonomous

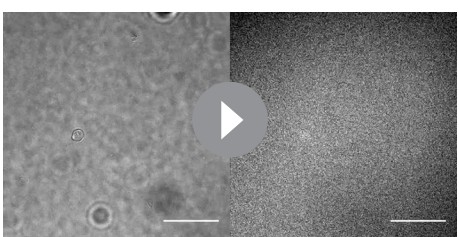

**Video 2.** Isolated segmentation clock cells oscillate in vitro. Field of view corresponding to cell in top row of *Figure 1—figure supplement 9*. Total duration = 6.2 hr; Time interval = 2.14 min, field size = 205 x 205 μm, Scale bar = 50 μm.

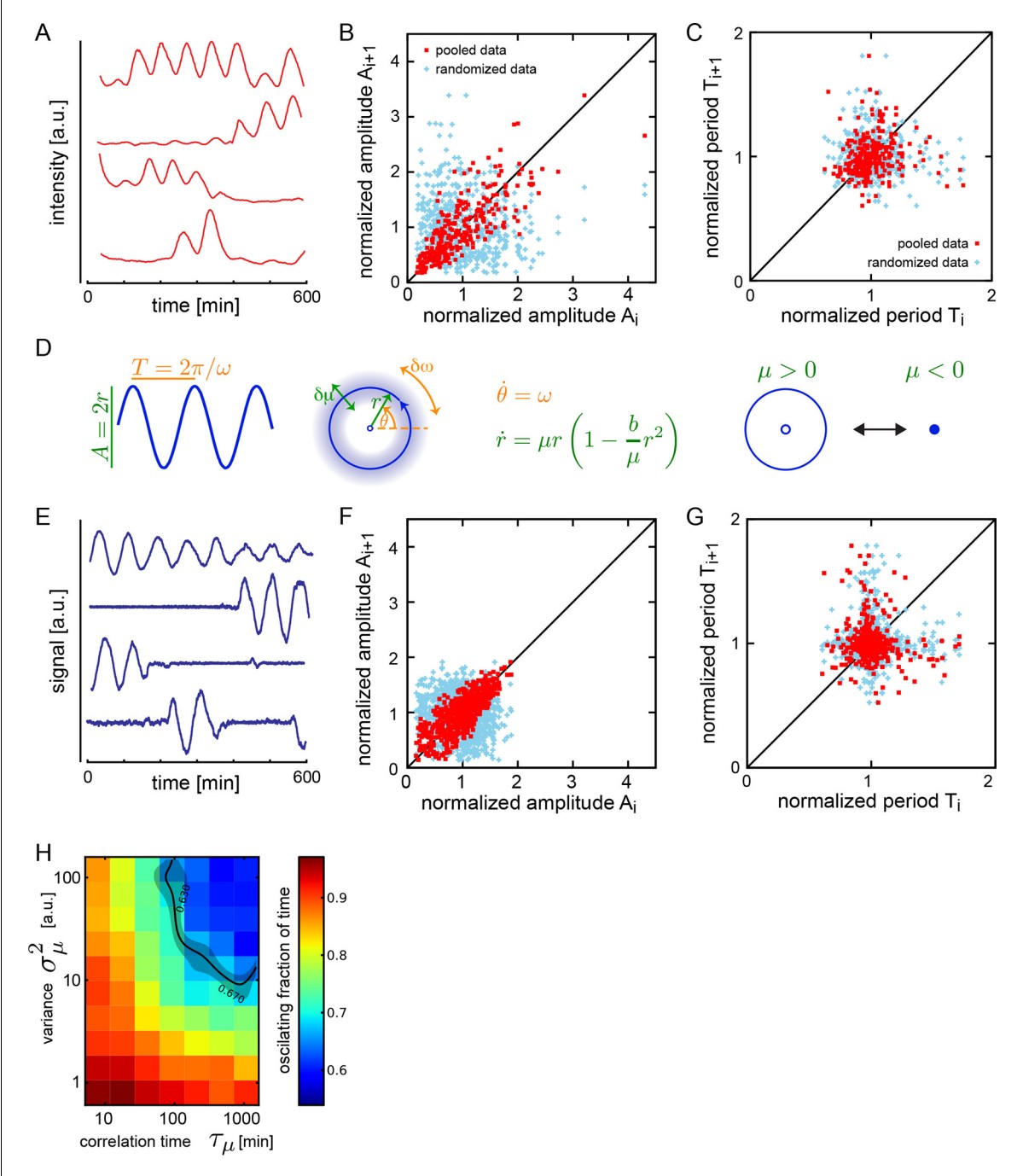

**Figure 2.** Dispersed low-density cells show a variety of behaviors compatible with slow amplitude fluctuations. (A) Representative background-subtracted traces displaying different oscillatory behaviors: persistent oscillations, oscillations that initiate, stop, or start and stop within the recording time of 600 min. (B) Amplitude correlations in successive cycles from the Fgf-treated low-density data set (645 cycles measured from 147 cells) are shown as red squares. Blue crosses show correlations from the same data set with pairs of peaks drawn at random from the same list. Amplitude values are normalized to the mean of the data set. (C) Period correlations in successive cycles from the Fgf-treated low-density data set are shown as red squares. Blue crosses show correlations from the same data set with period values drawn at random from the same list. Period is normalized to the mean of the data set. (D) Left: Scheme defining amplitude and period and corresponding limit cycle illustrating fluctuations in $\mu$ and $\omega$, which are parameters controlling amplitude and frequency, respectively. Middle: equation of the generic Stuart-Landau oscillator model, which describes the time evolution of phase $\theta$ and amplitude $r$ of the oscillator. Right: illustration of the Hopf bifurcation showing how the limit cycle (blue circle) collapses and becomes a fixed point (blue dot) as $\mu$ changes from positive to negative values. (E) Simulated traces generated with the Stuart-Landau model with colored noise in parameter $\mu$ and white noise in the oscillator variables, showing behaviors corresponding to those observed in the data, compare to

*Figure 2 continued on next page*

*Figure 2 continued*

panel A. (**F**) Amplitude correlations in successive cycles from the simulated oscillator are shown in red squares. Blue crosses show correlations from the same data set with pairs of peaks drawn at random. (**G**) Period correlations in successive cycles from the simulated oscillator shown in red squares. Blue crosses show correlations from the same data set with inter-peak-intervals from pairs of peaks drawn at random. (**H**) Heat plot of the fraction of time spent oscillating as measured by number of peaks occurring over time given the median period observed in the synthetic data, as the variance and correlation time of colored noise fluctuations in $\mu$ vary. Oscillating fraction of time for the Fgf-treated low-density data set (***Figure 1—figure supplement 5***; ***Figure 1—source data 3***) would be found in the shaded region of the heat plot.

The following figure supplements are available for figure 2:

**Figure supplement 1.** Statistics of amplitude and period correlations.

**Figure supplement 2.** Numerical simulations.

**Figure supplement 3.** Both additive noise and color noise are necessary for the theory to describe the observed fluctuations.

oscillators, are therefore not supported by our results. In addition, in nearly all cases the period of individual cells is longer than that of the tissue, indicating a role for tissue-level processes in setting the period of segmentation.

## Heterogeneity in the population of oscillating cells

The oscillatory signal we observe from individual cells is reporting the state of a pace-making circuit component (***Schröter et al., 2012***). A remarkable feature of these oscillatory signals is their variability between cells in the population (***Figure 1—figure supplement 5***). We observe a spectrum of behaviors in the low-density data set including cells that start or stop oscillating during the experiment, cells that start and then stop, and stuttering rhythms where cycles are missed (***Figure 2A***). Plotting amplitude and period of consecutive cycles from the whole low-density data set indicate that amplitude displays a slow variation revealed by correlations (***Figure 2B***), while period does not show any appreciable correlation at this timescale (***Figure 2C***; ***Figure 2—figure supplement 1A–C***). We did not find a correlation between the period and amplitude of each cycle (***Figure 2—figure supplement 1D***).

We adopted a theoretical approach to better understand these observations. We chose a generic Stuart-Landau (SL) model that describes the phase $\theta$ and amplitude $r$ of an oscillator in the vicinity of a supercritical Hopf bifurcation (***Figure 2D***) (***Strogatz, 1994***). In this description, the amplitude of oscillations grows with the square of the distance to the bifurcation. Existing genetic regulatory network models possess supercritical Hopf bifurcations (***Verdugo and Rand, 2008a***; ***2008b***; ***Wu and Eshete, 2011***), though the topology and parameter values of the pace-making circuit remain unclear (***Schröter et al., 2012***; ***Uriu et al., 2009***; ***Lewis, 2003***; ***Trofka et al., 2012***). The SL description allows us to examine the effects of noise strength and correlation time on frequency and amplitude, neatly separated and in combination (***Supplementary file 1***).

Amplitude fluctuations observed in the data occur on a timescale that is similar to that of the oscillator period, and could be the result of global changes in the cell state. Slow fluctuations in gene expression and signaling systems have been reported in a variety of systems (***Süel et al., 2006***; ***Sigal et al., 2006***; ***Huang, 2009***; ***Chang et al., 2008***; ***Albeck et al., 2013***; ***Aoki et al., 2013***). We introduced slow fluctuations in the parameter $\mu$ that controls the amplitude of oscillators in the theory. Motivated by the lack of correlation between period and amplitude in the data (***Figure 2—figure supplement 1D–I***), we set the coupling between these processes to zero. Slow amplitude fluctuations can drive the oscillators in and out of the oscillatory state (***Figure 2E***; ***Figure 2—figure supplement 2***), and introduce correlations in the amplitude of consecutive cycles that are comparable to the experimental data (***Figure 2F***). Interestingly, the trend to higher amplitude variance at higher amplitude values, and the existence of a low occurrence of high relative amplitude cycles are not captured by the theory. To describe period fluctuations and their weak correlation observed in the data we introduced an additive white noise in the variables of the oscillator (***Figure 2C,G***) (***Supplementary file 1***).

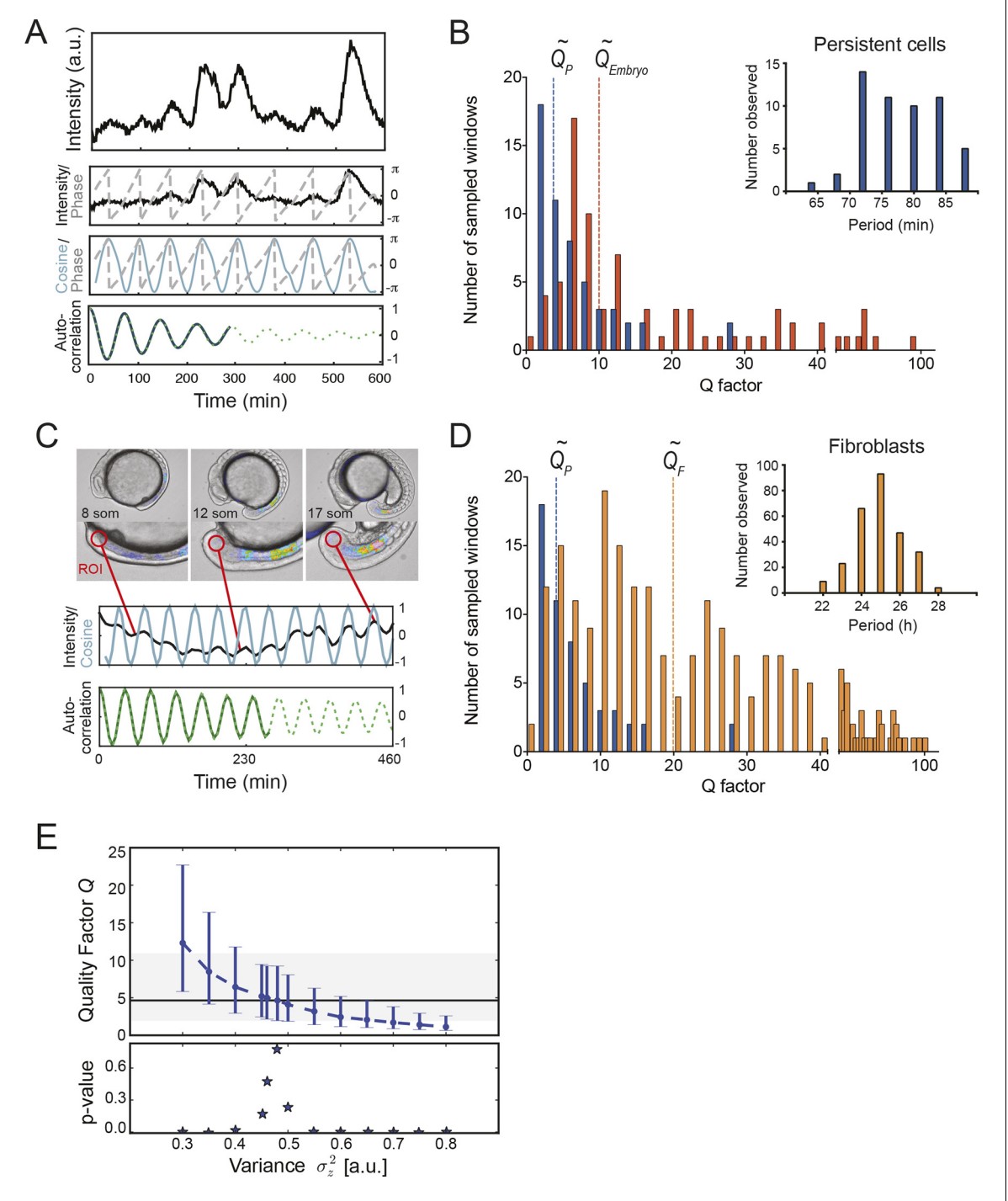

**Figure 3.** Precision of persistent segmentation clock oscillators (**A**) Quality factor workflow for time series analysis for an example persistent oscillator. Sub-panel 1: Background-subtracted intensity over time trace from a single tailbud cell (black) with phase (gray). Sub-panel 2: Wavelet transform of the intensity trace with cosine (light blue) of the phase information (gray). Sub-panel 3: Autocorrelation of the phase trace and fit (green) of the decay (for details see *Supplementary file 1*). The period of the autocorrelation divided by its correlation time is the quality factor plotted in B for each cell (blue). (**B**) Distribution of quality factors $Q_P$ for persistent segmentation clock oscillators (blue; range 1–28, median 4.6 ± 5.8) compared to quality factors $Q_E$ for the oscillating tailbud tissue in the embryo (red; range 1–117, median 10 ± 21). To compare between time series of different lengths we used sampling windows to calculate the quality factors, see theoretical supplement for details. Median values are indicated by dotted lines. Inset: Distribution of periods in single tailbud cells. (**C**) Estimation of tissue-level quality factor determined by measuring from an ROI placed over posterior PSM tissue in whole embryo timelapse of a single *Looping* embryo (*Soroldoni et al., 2014*). The intensity trace (black) and cosine (light blue) correspond to the average signal in the ROI over time. The period of the fit of the autocorrelation (green) divided by its correlation time is the quality factor plotted in B
*Figure 3 continued on next page*

*Figure 3 continued*

(red). (D) Distribution of quality factors for persistent segmentation clock oscillators (blue) replotted from B compared with the distribution of quality factors for circadian fibroblasts (orange; range 1–149, median 20 ± 27). Median values are indicated by dotted lines. Inset: Distribution of periods in circadian fibroblasts. (E) Precision decreases with increasing additive noise. Top panel, quality factor $Q$ vs. variance $\sigma_z^2$ of the additive noise, from numerical simulations (S30). Dots are the median value and error bars display the 68% confidence interval for 1000 stochastic simulations. Black line and shaded region indicates the median and the 68% confidence interval of persistent cells' oscillations. Bottom panel, p-value of a two-sample Kolmogorov–Smirnov test vs. variance $\sigma_z^2$. We test whether the persistent cells oscillations and the quality factors obtained from simulations come from the same distribution. In the absence of amplitude fluctuations $\sigma_\mu^2 = 0$, for $\sigma_z^2 = 0.486$ we have $Q = 4.6$ and a p-value of 0.78.

The following source data and figure supplement are available for figure 3:

**Source data 1.** Precision and period calculation for persistent segmentation clock oscillators.

**Source data 2.** Precision and period calculation for the tissue-level segmentation clock in the zebrafish embryo.

**Source data 3.** Precision and period calculation for persistent circadian clock oscillators.

**Figure supplement 1.** Quality factor value depends on length of time series.

Since oscillators move in and out of the oscillatory state, a key observable in both model and data is the fraction of the total time that a cell oscillates. We performed simulations over a range of values of correlation time and variance in $\mu$ and found a region in parameter space that corresponds to the behavior of isolated cells (*Figure 2H*). In contrast to these effects, changing the variance of the additive white noise did not affect the distribution of amplitudes, the correlations of amplitudes, or the fraction of the oscillating time (*Figure 2—figure supplement 3*).

While these results cannot rule out potential cell-type differences in the population, the theory is consistent with a population of cells having the same oscillatory mechanism, captured during different time windows of their dynamics. Importantly, it provides for the first time an observation of the longer timescales of noise in the autonomous oscillations of cells from the segmentation clock, regardless of its source.

## Precision of persistently oscillating cells

We next compared the precision of the most reliable of the cellular oscillators to the precision of collective oscillations in the intact embryo. From the 147 cells in the low-density data set, we selected those cells with persistent oscillatory behavior, defined as cells exhibiting sequential peaks over 80% of the recording time (n = 54 cells; *Figure 3—source data 1*). We used a wavelet transform to extract the phase of oscillatory signals (*Figure 3A*). We then evaluated precision by means of the quality factor $Q$, defined as the ratio of the decay time of the autocorrelation function and the period of oscillations (*Morelli and Jülicher, 2007*). All time series were sampled using time windows of equivalent lengths (6.5 cycles) (*Figure 3—figure supplement 1*); this procedure and the observed distribution of Q values are described in *Supplementary file 1*. We found that persistent cells had a mean period of 78.3 min (*Figure 3B*, inset). This is consistent with the period inferred from the inter-peak intervals measured from the entire low-density data set. We calculated a median quality factor $Q_P \sim 4$ from the persistent cells' oscillations (*Figure 3B*). Our analysis excluded dividing cells, and as cell division is thought to introduce phase noise in the time series (*Delaune et al., 2012*), the precision of dividing cells would be lower.

We compared the precision of persistent oscillators in vitro to the precision of the tailbud during an equivalent developmental time window using tissue-level oscillation data from embryos in *Soroldoni et al. (2014)*. We calculated a median quality factor $Q_E \sim 10$ in the embryo (*Figure 3B,C*; *Figure 3—source data 2*). This value indicates that the precision of the tissue level oscillations in vivo is a factor of 3 times higher than the typical persistent segmentation clock cell in vitro. This suggests that in the embryo, coupling may increase the precision of the individual cells by a similar amount. Given that there are many cells with lower Q factor than the median, including those that were too noisy to be included in the persistent oscillator set, this increase in precision should be considered as a lower bound.

To place the precision of persistent cells in context with another autonomous cellular oscillator, we analyzed the precision of the circadian clock using time series from single mammalian fibroblasts expressing a Period2-Luciferase reporter (*Leise et al., 2012*) and found these cells oscillated with a median quality factor $Q_F \sim 20$ (*Figure 3D*; *Figure 3—source data 3*). Thus, cells isolated from the zebrafish segmentation clock are less precise than the mouse circadian clock in single cells.

We investigated the precision of simulated oscillators, comparing it to the experimental data. Noise in the parameter $\mu$, which controls the amplitude and produced the heterogeneity of oscillator behavior discussed above, did not result in the observed range of quality factors (*Supplementary file 1*). In contrast, introducing white noise in the variables of the SL oscillator described the precision of the data (*Figure 2D*). This choice of noise was motivated by a lack of correlation in subsequent periods (*Figure 2C,G*). The experimentally observed precision of persistent cells from the segmentation clock was located within a restricted range of variance in this noise (*Figure 3E*).

Together these data demonstrate a role for collective processes in increasing the precision of the tissue-level segmentation clock above that of the individual isolated cells.

## Discussion

Our findings improve understanding of the segmentation clock at both single cell and tissue level. Single cell oscillators isolated from the zebrafish segmentation clock are autonomous. Nevertheless, tissue-level contributions aid in establishing the period and increasing the precision of segmentation.

In zebrafish, inhibiting the Delta-Notch pathway that mediates synchronization between neighboring oscillators results in slower segmentation (*Herrgen et al., 2010*). This change in period has been interpreted as the result of losing the collective effects of coupling with delays (*Herrgen et al., 2010*; *Morelli et al., 2009*). An untested prediction of this scenario is that an individual isolated tailbud cell should slow when removed from coupling within the tissue. Our experiments allowed us to compare the period of the low-density cells to that of the explanted *Looping* tailbud cultured under the same conditions.

We first noted that the period of the oscillations measured in tailbud explants (42.5 ± 11.4 min, mean ± SD) (*Figure 1—figure supplement 3*) was longer than the segmentation period in intact embryos of 27 ± 1 min (± SD) at 26°C (*Schröter et al., 2008*), a slowing of approximately 1.5-fold over the intact embryo. A comparable increase in period was not reported with explanted mouse tailbuds (*Masamizu et al., 2006*; *Lauschke et al., 2012*). However, a general developmental slowing of explanted zebrafish tissue has been previously reported (*Langenberg et al., 2003*). The reason for this is unknown, and likely involves chemical or mechanical differences in culture compared to the embryo. Recent studies of embryonic cell shape and migration have successfully utilized in vitro assays over minutes to tens-of-minutes time scales (*Diz-Muñoz et al., 2010*; *Maître et al., 2012*; *Ruprecht et al., 2015*), but the dynamics of longer-term zebrafish primary cell culture remains relatively unexplored (*Westerfield, 2000*). The differentiation of several lineages in primary cell culture appears to be slowed compared to the embryo, although this has not prevented the identification of relevant molecular regulatory mechanisms in this context (*Xu et al., 2013*; *Huang et al., 2012*). The zebrafish segmentation clock can maintain stable oscillatory output over a three-fold change in frequency due to temperature differences (*Schröter et al., 2008*), suggesting that its mechanism is robust to alterations in global growth conditions. Nevertheless, until the mechanism of this general slowing in vitro and its influence on the molecular and cellular processes within the segmentation clock are understood, we must remain circumspect in our interpretations.

The period inferred from the low-density data set was approximately two-fold longer than that of explanted tailbuds using the same time series analysis (*Figure 1—source data 1*). This supports the expectation that coupling with time delays between segmentation clock cells in the zebrafish leads to a decrease in the period of the synchronized population (*Herrgen et al., 2010*; *Morelli et al., 2009*). However, the magnitude of the difference is larger than anticipated from the segmentation period of intact embryos with reduced Delta-Notch signaling, where the effect of delayed coupling was estimated at 20% (*Herrgen et al., 2010*). This difference may be due to additional, as yet unknown coupling pathways in the tissue, and/or to the existence of signals in the tissue that alter the base period with which the cell can tick, and which are diluted by the low-density culture. Our own observations with a range of Fgf8b concentrations indicate that it does not affect period

(*Figure 1—figure supplement 8D*, and data not shown), suggesting that Fgf signaling is unlikely to be responsible. In this assay, rather than being instructive for the period of oscillations (*Ishimatsu et al., 2010*), Fgf appears to be permissive for the oscillatory state (*Dubrulle et al., 2001*).

In summary, our results demonstrate that individual cells have a longer period than the period in the tissue. Thus, they provide independent support for a role for collective processes in determining the period of the tissue-level segmentation clock.

A striking finding of our studies was the heterogeneity of oscillations across the population of cells. We propose that this heterogeneity can be described as the trajectories of self-sustained oscillators in the vicinity of a Hopf bifurcation. In this scenario, excursions across the bifurcation stop and start cycling behavior, but the underlying oscillatory mechanism remains. Fgf signalling appears to push the oscillators above this bifurcation, and the characteristic longer-timescales in the amplitude noise that we have observed may come from the inherent dynamics of the Erk network downstream of the Fgf receptor (*Aoki et al., 2013*; *Albeck et al., 2013*). One feature of oscillators in the vicinity of a bifurcation is that they may be more readily synchronized by coupling (*Gonze et al., 2005*; *Barral et al., 2010*). Individual cells from the mouse segmentation clock are also noisy, but the contribution of cell-cell signaling to maintenance of oscillations, either by contact or through diffusible factors, remains unclear (*Masamizu et al., 2006*). It is possible that in the mouse intercellular signaling may be required for sustaining oscillations, as has been observed for Delta-Notch signaling in neural progenitors (*Imayoshi et al., 2013*). Striking the optimal balance between autonomous and collectively-maintained oscillations could be a tune-able evolutionary strategy in developing multicellular systems. The relationship between cell intrinsic circuits, local cell communication and tissue-level patterns will be important for understanding development, as well as engineering strategies for synthetic cellular systems (*Sprinzak et al., 2010*; *Matsuda et al., 2015*).

## Materials and methods

### Explanted tailbud culture

The *Looping* zebrafish line expresses a fusion of the Her1 protein and YFP driven by the endogenous *her1* regulatory elements contained in a BAC transgene (*Soroldoni et al., 2014*). For tailbud explant cultures, intact posterior tissue including PSM and tailbud was dissected from transgenic zebrafish embryos between 5 and 8 somite stage. The ectodermal tissue layer was removed from the explant and discarded. Tailbud pieces were dissected away from the remaining PSM by making a lateral cut across the explant, just posterior to the base of the notochord and Kupffer's vesicle (*Figure 1A*). Explants were then transferred to fibronectin-coated glass bottom 35-mm petri dishes (MatTek, Ashland, Massachusetts) and maintained in a small volume of L15 medium (Sigma, St. Louis, Missouri) with 10% fetal bovine serum (Invitrogen, Waltham, Massachusetts) during imaging.

### Tailbud cell dispersals

Cultures of tailbud cells were generated for in vitro imaging as described in (*Webb et al., 2014*). Briefly, tailbud explants were generated as described above. For each independent replicate, multiple tailbud pieces (each containing ~1000 cells) were pooled and incubated in trypsin/EDTA (Sigma) for 20 min at room temperature. To quench trypsin, tailbud cells were dispersed into a small volume of L15 medium (Sigma) with 10% fetal bovine serum (Invitrogen) by pipetting, then plated onto a fibronectin-coated glass bottom 35-mm petri dish (MatTek) and allowed to adhere for 20 min Additional L15 medium containing 10% serum ± mouse Fgf8b (75 ng/uL; R&D Systems, Minneapolis, Minnesota) was added to the culture prior to imaging.

For the 96-well plate format, serial dilutions of the cell suspension to a final volume of 2 microliters were plated into individual wells. Again, additional L15 medium with 10% serum and Fgf8b was added to each well prior to imaging. Using this strategy, about 50% of plated wells have a single cell.

### Long-term timlapse imaginge

Imaging was performed as described in *Webb et al. (2014)*. Briefly, transmitted light, YFP and mCherry images were acquired using an EM-CCD camera (Andor xIOn 888, Northern Ireland) fixed to an inverted widefield microscope with a 40x lens (Axiovert 200M; NeoFluor 40x, NA 0.75, Zeiss,

Germany). The temperature of the imaging dish was maintained at 26°C using a Warner heating chamber (Harvard Apparatus, Cambridge, Massachusetts). Using iQ2 software, we acquired multiple fields within the dish over a 2-min interval and a 10-hr recording time.

## Image and time series analysis

In each recorded field, we counted the total number of cells that were YFP+ out of all viable cells ($n$ = 321 out of 547, 59%). We first disqualified from analysis YFP$^+$ cells that moved out of the field (4%), came into contact with another cell or divided (14%), or were not viable at the end of the 10-hr recording time (7%). The remaining cells (34%) were tracked in transmitted light images using a region of interest (ROI) tool in Fiji (ImageJ, NIH). Average and maximum intensities across the ROI were then measured and interpolated across images with a customized plug-in (ROI interpolator [*Soroldoni et al., 2014*)]). Before peak detection, any low frequency trends in the baseline of these raw intensity traces were removed by subtracting a local background obtained by measuring signal next to each cell (*Figure 1—figure supplement 2A*).

## Peak finding and period/amplitude statistics

For each background subtracted cell trace, we detected peaks using a custom Matlab script that uses the findpeaks function. The algorithm first smoothens the time series and then detects local maxima (*Figure 1—figure supplement 2A*). Local minima are then found between pairs of maxima using the function min restricted to the time interval between successive peaks. Peaks are subsequently discarded if they are smaller than 0.1 times the dynamic range of the time series, or if they are too close (less than 40 min) to the beginning or end of the time series, see the example trace in *Figure 1—figure supplement 2A*.

Period is estimated as the time interval between consecutive peaks (*Figure 1—figure supplement 2B*). Period estimates are discarded if their value is larger that 140 min, interpreting these events as elapsed time between disconnected peaks. Amplitude is defined as the average of the difference between a peak's height and its two adjacent minima, to take into account signal drifting during one cycle.

## Generation and validation of Ntla and Tbx16 antibodies

Monoclonal antibodies were generated to the Ntla protein, the zebrafish *T/Brachyury* homolog, and to Tbx16, the product of the *spadetail* locus. 8 µg of Ntla (amino acids 1–261) or Tbx16 peptide (amino acids 232–405) fused to GST was injected into Balb/c mice; sera were screened via ELISA. Each antiserum with a positive signal was further tested for tissue-specific binding in 15-somite stage wild-type and mutant or morpholino-injected embryos. Hybridoma cell lines were produced from one mouse; antibodies were purified from the supernatants. The antibody with highest signal-to-noise ratio was used for experiments (Ntla, clone D18-4, IgG1; Tbx16, clone C24-1, IgG2a). The validation of these antibodies is shown in *Figure 1—figure supplement 6*.

## Immunocytochemistry for PSM markers

Cell dispersals from the same cell suspension used for time-lapse imaging (L15 medium, plus 10% serum and Fgf8b) were cultured separately on Conconavalin-A (Sigma) coated glass bottom dishes (MatTek) and maintained in the incubator at 28°C. These cells were fixed in 4% paraformaldehyde (Sigma) after 5 hr in culture overnight at 4°C. Prior to staining, cells were washed 4 × 5 min in PBS at room temperature and then blocked for 1 hr at room temperature in a PBS solution containing 1% BSA and 0.1% Triton (Sigma). For staining, cells were incubated with monoclonal antibodies for Ntla (D18, IgG1) and Tbx16 (C24, IgG2a) (1:5000) overnight at 4°C. Primary antibody was removed and the cultures were washed 3 × 20 min in PSM at room temperature. Cultures were incubated in secondary antibodies GFP-anti-IgG1 and Cy5-anti-IgG2a (1:500, Molecular Probes, Eugene, Oregon) for 2 hr at room temperature, prior to DAPI staining and final washes (*Figure 1—figure supplement 7*).

## Circadian fibroblast data set

These data were generated and published in *Leise et al. (2012)* and kindly donated for analysis.

## Precision measurements

We use wavelet transforms to generate a phase time series from the raw traces. We compute the autocorrelation function of this phase and fit it to obtain an estimate of the period $T$ and correlation time $t_c$, which allows us to compute the quality factor $Q = t_c/T$ (*Supplementary file 1*).

## Stuart-Landau oscillator model

The observed behavior can be described using a Stuart-Landau model, which captures the generic behavior of an oscillator close to a supercritical Hopf bifurcation that gives rise to sustained, limit cycle oscillations. We introduce slow fluctuations in the parameter that controls the distance to the bifurcation to capture amplitude fluctuations, and white noise in the variables of the oscillator to capture frequency fluctuations (*Supplementary file 1*).

## Acknowledgements

The authors would like to thank David Welsh for providing the fibroblast data set, Ravi Desai for assistance with widefield microscopy, and Ulrike Schulze for assistance with confocal microscopy, Daniele Soroldoni for general wisdom and abilities to discuss science while consuming alcohol, David Dreschsel and the MPI-CBG Protein Production Facility, Patrick Keller and the MPI-CBG Antibody Facility, the Oates lab for comments and discussion, and Philip Murray, Kim Dale, Saul Ares, Stephanie Taylor, Hanspeter Hertzel, Ravi Desai, Laurel Rohde and Caren Norden for comments on a draft of the manuscript.

## Additional information

### Competing interests

FJ: Reviewing editor, *eLife*. The other authors declare that no competing interests exist.

### Funding

| Funder | Grant reference number | Author |
| --- | --- | --- |
| European Research Council | 207634 | Alexis B Webb<br>Guillaume Valentin<br>Luis G Morelli<br>Andrew C Oates |
| Medical Research Council | MC_UP_1202/3 | Alexis B Webb<br>Guillaume Valentin<br>Andrew C Oates |
| Wellcome Trust | WT098025MA | Alexis B Webb<br>Guillaume Valentin<br>Andrew C Oates |
| European Molecular Biology Organization | ALTF 1565-2010 | Alexis B Webb |
| National Science Foundation | OISE 1064530 | Alexis B Webb |
| Agencia Nacional de Promoción Científica y Tecnológica | PICT 2012 1954, 2013 1301 | Iván M Lengyel<br>Luis G Morelli |
| Max-Planck-Gesellschaft | | Alexis B Webb<br>David J Jörg<br>Guillaume Valentin<br>Frank Jülicher<br>Luis G Morelli<br>Andrew C Oates |

The funders had no role in study design, data collection and interpretation, or the decision to submit the work for publication.

### Author contributions

ABW, Conception and design, Acquisition of data, Analysis and interpretation of data, Drafting or revising the article; IML, DJJ, LGM, Designed and executed the theoretical analyses, Analysis and

interpretation of data, Drafting or revising the article; GV, Acquisition of data, Analysis and interpretation of data, Contributed unpublished essential data or reagents; FJ, Design and execution of theory, Analysis and interpretation of data, Drafting or revising the article; ACO, Conception and design, Analysis and interpretation of data, Drafting or revising the article

## Author ORCIDs

Frank Jülicher, http://orcid.org/0000-0003-4731-9185
Andrew C Oates, http://orcid.org/0000-0002-3015-3978

## Ethics

Animal experimentation: Zebrafish experimentation was carried out in strict accordance with the ethics and regulations of the Saxonian Ministry of the Environment and Agriculture in Germany under licence Az. 74-9165.40-9-2001, and the Home Office in the United Kingdom under project licence PPL No. 70/7675.

## Additional files

**Supplementary files**

• Supplementary file 1. Theory supplement including Stuart-Landau theory with slow fluctuations, quantification of noisy oscillations, and numerical methods.

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
