## [Decision Letter]

[Editors’ note: this article was originally rejected after discussions between the
reviewers, but the authors were invited to resubmit after an appeal against the
decision.]

Thank you for choosing to send your work entitled "Persistence, period and
precision of autonomous cellular oscillators from the zebrafish segmentation clock"
for consideration at *eLife*. Your full submission has been evaluated by
Janet Rossant (Senior editor), Tanya Whitfield (Reviewing editor and reviewer), and two
additional peer reviewers, and the decision was reached after discussions between these
individuals. Based on our discussions and the individual reviews below, we regret to
inform you that your work will not be considered further for publication in
*eLife*.

The reviewers agreed that the study is interesting and has been done carefully, but they
had significant concerns over the fit of the experimental data to the model, and the
extent to which the analysis provides a true advance in our understanding of
somitogenesis in vivo. Although the study uses a larger number of cells than in previous
work on similar systems, it was felt that the main findings are unsurprising and largely
confirm those of earlier studies. It was also felt that, should a link to in vivo data
be made (as suggested by Reviewer 3), this was also likely to confirm work that has
already been done, and so would be unlikely to add further understanding. The full
reviews are appended below.

*Reviewer #1:*

This study tests the proposal that the zebrafish somite segmentation clock is
characterised by autonomous cellular oscillators, which have been proposed in other
studies to be present and coupled by Notch signalling. This question has been tackled
before in other species (chick, mouse), although in these previous studies, a very few
dissociated cells were studied as part of a cell suspension (Masamizu et al.) or were
pooled for analysis (Maroto et al.). The current study provides an advance over these
previous studies, as the authors have examined much larger numbers of dissociated cells
within a low density suspension than in previous studies, enabling a much more thorough
quantitative and statistical analysis of the data. The authors also examine a small
number of cells that have been isolated completely, and show (for a small number of
cells) that oscillations can occur autonomously under these conditions.

Overall, the study appears careful and thorough and will be an important addition to the
literature in this field. The experimental detail is sometimes a little sparse and more
information could be added for clarification – see specific comments below. In addition,
the figure legends need careful revision, as there is confusing mislabelling in several
places.

Specific comments:

1) The period of individual cell oscillations at 26°C (81 min) is much longer that the
value that is stated to be the period of trunk segmentation in intact embryos (27 min),
measured tissue explants from the Looping Tg strain (55 min) and the authors'
predictions (40 min) (subsection "Inferred period of segmentation clock cells in
vitro”). This needs further explanation and discussion. What is the genetic background
of the Tg strain? This is not described in the Materials and methods. The authors should
include a measurement of the period of trunk segmentation in intact Looping Tg strain
embryos as well as the explants, rather than merely comparing to previously published
work (albeit from the same group, but done in a different laboratory etc.).

2) Little information is given about the health and other characterization of the cells
in the low density suspensions. In the videos, they appear to show a lot of blebbing
activity – is this expected? Do they retain other characteristics (e.g. gene expression)
of the PSM and are they able to divide? It would be helpful to have some more
information about the general behaviour of these cells under the conditions of the
experiment.

3) What happened to the cells that touched each other? Do pairs of cells in contact with
one another synchronise their signalling, or oscillate with a shorter period? Is there a
community effect, and what is the minimum number of cells in a group that is sufficient
to maintain faster and synchronized oscillations?

4) Fgf8b has been added to the culture medium to mimic conditions in the tailbud. What
happens in absence of Fgf8b? Are there oscillations at all? Or damped oscillations? or
oscillations with even longer periods?

5) The legend to Figure 1—figure supplement 1
is not sufficient to understand the figure. There is both a red and a grey line on each
panel, but the legend confusingly only refers to a black line. (What is the grey line? –
Background levels? – This should be stated somewhere.) I presume that the grouping of
each set of traces refers to the independent replicates, but this is not explained
anywhere. The 'smoothed' traces should be labelled as such on the figure. Also, the
colour code is not correct – the smoothed traces are in red with blue circles to
indicate peaks and troughs, not in blue with red triangles as indicated in the legend.
Likewise, in Figure 1—figure supplement 2 and
legend, the peak finding on the traces is actually labelled with red triangles, not blue
as indicated in the legend; filtering appears to be labelled with blue triangles, not
red as indicated. This lack of attention to detail has made it difficult and
time-consuming to review the manuscript.

6) Figure 1—figure supplement 6 – these traces
show timecourses for 10 individual isolated cells. Have these data been subjected to the
same smoothing, peak and trough analysis etc. as the data from the low density set? Do
they behave in a similar way?

*Reviewer #2:*

The manuscript presents a very worthwhile data set, detailing time-course expression
profiles for a Her1 reporter with high temporal resolution. These show a high degree of
variability, and a range of analyses are performed to extract phase and amplitude data.
The authors go on to compare these summary measures to those obtained from a generic
model of a system close to a supercritical Hopf bifurcation (at the boundary between
decaying and sustained oscillations). They show that reasonable (though not perfect)
agreement between the two sets of measures (experimental and simulated) can be achieved
for an appropriate set of model parameters.

These results are interesting in their own right, as a careful investigation of the
nature of the time-courses obtained from the cultured tailbud cells. I'm less convinced
that they say anything particularly important about somitogenesis in zebrafish. My
primary concern here is that the Her1 expression observed in the low-density cultured
cells is so distinct to that observed in tailbud cells in an intact embryo. The period
of oscillation in vivo is 27min; the mean period extracted from the time-courses of the
cultured cells is around three times as long as this (ca. 78min). The two sets of
oscillations thus represent markedly different dynamics. The long-period noisy
oscillations observed in cultured cells may (or may not) be due to a dynamical system
close to a Hopf bifurcation, but if they are, it is hard to see how this relates to a
dynamical system based on the same components oscillating in a tissue with almost three
times the frequency! The key observation in this regard is that the somitogenesis period
in explanted PSM is around 55min, rather than the 27min in the embryo at the same
temperature. This is reported in this manuscript without any real discussion. But surely
this is a striking and potentially important result.

My overall assessment is that the data set and data analysis are high quality and worthy
of publication. The potential fit to a very generic model is also interesting, but I'm
not convinced that the fit is actually that good. Given the large discrepancy between
the cultures and the embryo, I'm not convinced that the data have much to say about
somitogenesis.

Substantive concerns:

1) Fgf8b was added to the culture medium "to mimic the signaling environment of the
tailbud". Was this necessary to see oscillations in Her1-YFP levels? What happens
if it is not added?

2) Did cells in culture never divide? This is not mentioned (other reasons for
discounting time-courses are mentioned). Cells in the tailbud typically undergo a single
division at around 15-16hpf (see Figure 1 in
Bouldin et al. (2014). Genes & Development, 28, 384-395.) It seems that this
division should fall within the time window of the current study. Are the culture
conditions blocking this division, and if so, might that be expected to affect the
oscillations?

3) The authors state (subsection “Heterogeneity in the population of oscillating cells”)
that the simulated data are "in good agreement with the experimental data (Figure 2)" However, Figure 2 don't match very well, as noted by the authors
themselves in the next sentence. There is clearly more overall variability in amplitude
in the real data (2B) than in the simulated data (2F), though the correlation of
neighbouring peak amplitudes is similar. So the data show similar short-time coherence
of amplitude to the model, but significantly greater variability in amplitude
overall.

4) The data do appear to support the idea that the main variability in the oscillations
is in the amplitude rather than in the period (Figure 2). The authors use this finding to focus their attention on a S-L model
with q=0 (so, the frequency is independent of the amplitude). This restricts the type of
supercritical Hopf bifurcation that could underlie the observed oscillations (the normal
form does not require q=0). Does this restrict the possible molecular mechanisms
underlying the observed Her1 oscillations? In particular, do noisy negative feedback
oscillators behave in this way? It is my understanding that stochastic simulations of
negative feedback circuits (like the ones alluded to by the authors) that exhibit
sub-Hopf stochastic resonance do show amplitude-dependence of the period (increased
amplitude leads to increased period). Furthermore, the authors have previously shown
that the use of the Her1-Venus transgene increases the period of oscillations. Is this
likely to be due to the overall increase in expression level of Her1 in the Looping
line? The amplitude-dependence of period (or lack of it) is potentially important for
the tissue-wide oscillations observed in the PSM, as the value and distribution of q is
an important factor in the ability of oscillators to synchronise.

*Reviewer #3:*

Webb et al. describe segmentation clock oscillations of individual dissociated zebrafish
tail bud cells. They find that oscillations are present in vitro (as expected), but that
the period is longer than in vivo (as previously postulated but shown here for the first
time). A noise model can account for the degeneration of oscillations.

The data is interesting for zebrafish researchers but I am not convinced that it goes
beyond previous studies in chick and mouse.

It seems straightforward to compare the in vitro oscillations to the in vivo
oscillations in Notch signaling mutants. This experiment would test if the differences
in vitro can be explained by the absence of Notch signaling.

[Editors’ note: what now follows is the decision letter after the authors submitted for
further consideration.]

Thank you for resubmitting your work entitled "Persistence, period and precision of
autonomous cellular oscillators from the zebrafish segmentation clock" for further
consideration at *eLife*. Your revised article has been evaluated by
Janet Rossant (Senior editor) and three reviewers, one of whom, Tanya Whitfield, is a
member of our Board of Reviewing Editors.

The reviewers agree that the manuscript has been improved and that the study is a
significant advance over previous work. Although several unanswered interesting
questions remain, the study will be valuable in stimulating discussion and research in
this area.

However, there are also some concerns. The first, outlined by Reviewer 2, is that the
oscillations you describe in single cells do not necessarily represent the 'segmentation
clock' itself. This may require some re-wording or further discussion in the
manuscript.

The second concern, highlighted by Reviewer 1, relates to the cell division rate in your
cultures. This is perplexing, as the video that you show illustrates a mitotic index an
order of magnitude higher than the value you state for the study overall (5/18 cells
dividing in this field of view = 28%, whereas you state that the rate overall is only
2%). As your cells are drawn from a dividing progenitor population, and Fgf8 (a mitogen)
has been added to the culture, it would be expected that the cells continue to divide,
and it is reassuring to see them do so. However, the discrepancy between this and your
description makes it difficult to have confidence in the numbers of dividing cells
stated in the manuscript. Since this has implications for interpretation of the dataset
as a whole, it is important that this is addressed.

*Reviewer #1:*

Firstly, and also with respect, I would like to clarify that I did not state that the
previous manuscript was 'only a technical' advance over previous work; in fact, this
part of my review was a positive comment in support of the paper.

The authors have addressed all the points that I raised, most of them very
satisfactorily. I think the argument is clearer and that the additional data (e.g.
maintenance of marker expression) help with the overall interpretation.

My one remaining concern relates to the issue of dividing cells in the low-density
dataset. It is helpful that the authors have given more detail on cell divisions in a
source file, and I appreciate the argument that the focus of the study is necessarily on
the single cells. However, more explicit information is still needed here, especially
for Video 1. The legend to Video 1 indicates that it corresponds to the
field of view for the non-dividing cell in Figure 1. It would help to mark the cell of interest (e.g. with a ring or arrowhead)
at the start of the video. Otherwise, the text in the fifth paragraph of the subsection
“Oscillations in isolated segmentation clock cells in vitro”, leads the reader to think
that Video 1 is illustrating a whole field of
cells that do not divide.

More importantly, however, in addition to the non-dividing cell in the lower left, Video 1 also includes no fewer than five clear
examples of cell divisions, four of them YFP+ (at 00.26 (top right and lower right),
00.30 (middle left), 00.33 (extreme top right) and 00.41 (bottom right)). I find this
very surprising, given the source data file ([Supplementary-material SD2-data]), which documents only 9 instances of
cell division in the entire dataset from a total of 29 fields of view, which would
suggest that cell division events are rare. The observation of five division events in a
single field of view also suggests that Video 1 is not representative of the dataset as a whole, and is directly
contradictory to the statements in the rebuttal that 'it seems that culture conditions
are inhibiting division'; 'because of the very low rate of cell division in the culture'
etc. In short, I am not confident that the authors have measured the rate of cell
division accurately in their cultures. Further explanation is required here.

As the authors have pointed out in their rebuttal (although I could not find this in the
manuscript), the pairs of daughter cells remain attached to one another. As far as I can
see, there does not appear to be a consistent timing of division relative to the phase
of the oscillation between the three examples, or maintenance of synchrony between
daughters within a pair, although oscillation in at least one daughter can clearly
continue after division. For example, after the division at 00.30, the daughter on the
left of the pair has a very clear YFP peak at 00.39-00.40, while its sister on the right
has very low YFP signal at the same time point. Since the dataset captures this very
interesting information, it deserves some comment, even if a full analysis is beyond the
scope of the current study.

For cells that did divide (including the four YFP+ cells shown in Video 1), were oscillations measured up to the point of division,
or were these cells eliminated from the analysis altogether? The text is not altogether
clear on this point.

*Reviewer #2:*

I continue to believe that the data described in this manuscript are interesting and
valuable, and I think it is very positive that the authors are making the raw data
available to allow others to analyse them (although I maintain that the authors have
done a good job of this already).

The authors have addressed my queries and concerns. The additional data, and provision
of raw data, make it easier to assess the validity of the statements made in the
manuscript.

I have one remaining niggle with the conclusions drawn from the data and the
corresponding nomenclature employed by the authors. The cells are derived from the
tailbud and maintain tailbud markers (new data). The cells certainly exhibit
oscillations, which have features that are consistent with an underlying dynamical
system that is close to a Hopf bifurcation. So it is fair to say that the cells are
oscillators. However, I remain to be convinced that these facts are sufficient to call
the cells "single zebrafish segmentation clock cells" (e.g. sixth paragraph of
the Introduction and many other places). The segmentation clock is surely a coordinated
tissue-wide oscillatory operating in the tailbud/PSM. A parsimonious (though not
necessary) explanation of the findings reported is that the segmentation clock results
from the interaction of some noisy cell-autonomous genetic oscillators. But that is not
sufficient to call the oscillations in the single cells in vitro a "segmentation
clock".

I accept the argument that the period of the in vitro segmentation clock can exhibit
marked temperature-dependence. But that does not lessen the difficulty of arguing that
if a cell-autonomous oscillation in tailbud-derived cells is borderline-Hopf with a
period of ca. 80min, then the in vivo clock, with a period of ca. 27min, has something
to do with borderline-Hopf dynamics. I maintain that I do not think it is easy to
explain how an oscillatory mechanism can be close to a Hopf bifurcation at two radically
different periods.

It may be that the mechanisms underlying the observed in vitro cell-autonomous
oscillations and the in vivo segmentation clock are closely related (and it is probably
true), but that does not mean that their dynamics are similar. Just because cells can
oscillate in two different conditions does not mean that one can say that the
oscillations are two different versions of the same thing.

In summary, I like this study. I think the in vitro oscillations are interesting and the
analysis is powerful. I just don't like the idea of calling these oscillations a
segmentation clock – that is what operates in the embryo.

*Reviewer #3:*

Webb et al. have revised their manuscript to more clearly explain the limitations of
previous studies in concluding that cell-autonomous oscillations form the basis of the
segmentation clock. The differences between single cells, tissue explants and in vivo
tail buds are striking but unexplained. The paper is a nice technical advance but the
main conclusions were expected from previous studies, and the most interesting phenomena
of different periods in vitro and in vivo remains unaddressed.

---

## [Author Response]

[Editors’ note: the author responses to the first round of peer review follow.]

*Reviewer #1: This study tests the proposal that the zebrafish somite
segmentation clock is characterised by autonomous cellular oscillators, which have
been proposed in other studies to be present and coupled by Notch signalling. This
question has been tackled before in other species (chick, mouse), although in these
previous studies, a very few dissociated cells were studied as part of a cell
suspension (Masamizu et al.) or were pooled for analysis (Maroto et al.). The current
study provides an advance over these previous studies, as the authors have examined
much larger numbers of dissociated cells within a low density suspension than in
previous studies, enabling a much more thorough quantitative and statistical analysis
of the data. The authors also examine a small number of cells that have been isolated
completely, and show (for a small number of cells) that oscillations can occur
autonomously under these conditions.*

We would respectfully dispute the claim that our manuscript provides only a technical
advance over the papers of Masmizu and Maroto. Although these pioneering papers have
been cited in the primary and review literature as containing evidence for autonomous
oscillators in the mouse or in the chick, in fact, neither of these papers provides
direct evidence. Indeed, the authors of the papers have carefully refrained from making
any conclusions about this aspect, and based on the evidence in these papers, one could
just as easily conclude that cells from the chick and mouse segmentation clock cannot
autonomously sustain oscillations. We argue that this is a non-trivial problem with the
literature. The idea that segmentation clock cells are autonomous has been widely
assumed for the purposes of theoretical modeling; this assumption was first explicitly
stated in 2003 in three papers, two years before Maroto and three before Masamizu:

1) Lewis, J. (2003) Autoinhibition with transcriptional delay: a simple mechanism for
the zebrafish somitogenesis oscillator. Current Biology 13(16), 1398–1408;

2) Monk, N. A. M. (2003). Oscillatory expression of Hes1, p53, and NF-kappaB driven by
transcriptional time delays. Current Biology 13(16), 1409–1413;

3) Jensen, M. et al., (2003). Sustained oscillations and time delays in gene expression
of protein Hes1. FEBS Letters, 541(1-3), 176–177.,

It is possible that these experimental papers were selectively interpreted by the field
in the light of the earlier theoretical models, and the idea that this has somewhere
been shown experimentally appears now to be widely believed.

We take some space below to detail the evidence and claims of these two important
papers.

Maroto et al., 2005: In this study, cells were isolated from chick posterior PSM, then
dispersed and grown in suspension culture, and fixed at subsequent time points over an
interval spanning the formation of 2 somites (3 h). The fixed cells were spun down onto
a slide to allow in situ hybridization with mRNA to the cyclic gene Lunatic fringe
(Figure 3). The authors observed changes in
the percentage of cells expressing Lfng in these different time points, however, they
were not able to distinguish between noisy autonomous oscillators and stochastic
patterns of gene expression (p313): “Such fluctuating percentages are likely to
correspond either to a stochastic shut down of oscillating cyclic gene expression or to
asynchronous oscillations among cells of the same pool.” The authors highlighted the
need for real-time reporters to investigate whether PSM cells are sustained, autonomous
oscillators (p314): “The development of real-time imaging techniques to analyze cycling
gene expression at single cell resolution will be required to establish whether
oscillations are maintained or not in these cell cultures.”

Masamizu et al., 2006: In this study the first real-time reporter of the segmentation
clock, a luciferase reporter of Hes1 expression in mouse, allowed single PSM cells to be
observed in vitro. However, the arguments in this paper about oscillations and autonomy
need to be treated with caution. The authors first state (p1314) “It was previously
shown that expression of the chick homolog c-hairy1 oscillates even in dissected PSM
fragments, suggesting that this oscillator functions in a cell-autonomous manner (7,
19).” This is mistaken reasoning by the authors, since this result says nothing at all
about cell- autonomous function, only whether the PSM needs to be intact or needs a
neighboring tissue to oscillate. The result is equally consistent with cell-autonomous
and cell non-autonomous oscillations.

When introducing the PSM dispersal experiments, the authors write: “It was recently
shown that dissociated PSM cells also become out of synchrony (19).” But, this is a conclusion that the authors of reference 19
(Maroto et al.,) did not feel confident about, as discussed above. They continue:
“However, it is not clear whether each PSM cell has a stable oscillator but is reset at
various phases when dissociated or has an unstable oscillator, like fibroblasts. We thus
next examined Hes1 oscillation in dissociated PSM cells.” In this latter passage
Masamizu assume that the cells oscillate, and couch their description in these
terms.

The entire PSM except S0 was dispersed in 100% serum and cells were plated on poly-L
lysine glass bottomed wells. The density of plating was not stated, and it was not
reported whether cells were touching neighbors during the course of the recording. It is
not stated how many cells were imaged, but expression from only 3 cells was reported,
each showing 4 expression pulses with variable duration and steeply decreasing amplitude
(Figure 4 D, E). The original location of these cells in the PSM prior to dispersion is
not known, nor how frequent this behavior is amongst the cells in culture. A period was
given as 155 ± 6 min in the text, however, it was not stated how this was calculated. It
is difficult to see how this data supports the idea that mouse PSM cells are sustained
autonomous oscillators.

Importantly, Masamizu never use the phrase “cell autonomous” to describe their own data.
Our reading of Masamizu is that they have probably filmed rare events of PSM cells
rapidly damping out oscillations, in a cultured field of otherwise non-oscillatory
cells. Consistent with this interpretation is the absence of any study in the last 10
years using any of the various mouse reporter lines to show potential cell-autonomous
properties of the oscillators.

In summary, Masamizu’s study proposes that PSM cells may be “unstable oscillators”, and
highlights the role of inter-cellular signaling in maintaining and coordinating
oscillations in vivo. Reflecting this perspective, their mathematical model of the
process did not contain a sustained oscillator. Instead, the formalism is an example of
an excitable system where the expression pulses are initiated by fluctuations or by
signals from neighboring cells (Figure 5).

It is not our place here to be overly critical of Masamizu, as this paper was a landmark
in the field because of the live reporter. However, it is important to point out exactly
the strengths and limitations of their study, and to be clear about what sort of
evidence is there, what sort of conclusions can be reasonably drawn, and what remains an
open question. In our revised version, we have significantly modified the Introduction
to more carefully describe the data from these papers and their context in the
literature.

*Overall, the study appears careful and thorough and will be an important
addition to the literature in this field. The experimental detail is sometimes a
little sparse and more information could be added for clarification* –

*see specific comments below. In addition, the figure legends need careful
revision, as there is confusing mislabelling in several places. Specific
comments:*

*1) The period of individual cell oscillations at 26°C (81 min) is much longer
that the value that is stated to be the period of trunk segmentation in intact
embryos (27 min), measured tissue explants from the Looping Tg strain (55 min) and
the authors' predictions (40 min) (subsection "Inferred period of segmentation
clock cells in vitro”). This needs further explanation and discussion. What is the
genetic background of the Tg strain? This is not described in the Materials and
methods. The authors should include a measurement of the period of trunk segmentation
in intact Looping Tg strain embryos as well as the explants, rather than merely
comparing to previously published work (albeit from the same group, but done in a
different laboratory etc.).*

We agree that this aspect of the manuscript was confusing. In response to this comment
and to comment 2 and 4 below, and to very similar to comments by reviewer 2 (concerns 1
and 2), we decided to fundamentally reorganize the beginning of the Results section, and
to include new experiments. We replaced the original estimates of period differences
with a direct measurement of the difference between the period of the explanted tissue
and the separated cells. Importantly, we combine this with describing the step-wise
isolation of the cells, documenting the changes at each step from explanted tissue to
isolated cells. We hope that this better illustrates the connection between the
different samples, and also makes it clear exactly at which steps differences in the
period appear. Note that we cannot yet explain all the differences, but we hope the new
manuscript allows the reader to better understand what is known.

We include a new experiment in which we explanted and grew the tailbuds from Looping
embryos in similar culture conditions to the isolated cells. These explants show
persistent oscillations, but with a longer period than the intact embryo (Figure 1—figure supplement 3). We next describe
the dispersed individual cells from the tailbuds in identical conditions, and show that
most show very few cycles (median 2) with a twofold longer period before damping out
(Figure 1—figure supplement 4). Motivated by
the elevated level of FGF in the tailbud in vivo, we next describe the addition of Fgf8b
to the cultures. This does not further change the period, but dramatically increases the
number of cycles to a median of 5 (Figure 1—figure supplement 5). In addition, we have included a spreadsheet in which the raw
and background-subtracted data from all low-density oscillating cells is presented so
that other researchers may analyze this in detail ([Supplementary-material SD3-data]).

The results from each time series analysis are presented in table form in [Supplementary-material SD1-data]. Thus,
explanting slows the tissue period about 1.5 times, and dissociating the cells from the
tissue slows them about 2-fold. We do not understand the origin of the general slowing
in culture, but we comment on the fact that this has been previously noted by others,
and propose that some general property of culture may be responsible. We also point out
that primary culture studies for zebrafish is in its infancy in the literature,
particularly with respect to dynamics, and that obviously more work needs to be
done.

A slowing of individual cells due to separation from the tissue was predicted by
previous work, and we see this, but the observed magnitude is higher than expected. This
is actually quite exciting, and forces us to propose the existence of a second, as yet
unknown coupling system and/or a constitutive factor that elevates the period, but is
lost by diffusion when the cells are separated (but is obviously not FGF).

We have added text to introduce and explain the transgenic line, which was generated in
Soroldoni et al., (Science 345, 222–225, 2014), in a timely manner at the beginning of
the Results and information in the Materials and methods. The new Results subsection
(“Oscillations in isolated segmentation clock cells in vitro”) describes the step-wise
dissociation, the observed periods, and additional measures of the cells’ properties
(shape, division, gene expression etc.). We have removed the previous Results section
that focused on the calculation of differences in estimated period, and have instead
included a discussion of the period differences in the subsection “Heterogeneity in the
population of oscillating cells”.

Please also see our response to reviewer 2, below.

*2) Little information is given about the health and other characterization of
the cells in the low density suspensions. In the videos, they appear to show a lot of
blebbing activity* –

*is this expected? Do they retain other characteristics (e.g. gene expression) of
the PSM and are they able to divide? It would be helpful to have some more
information about the general behaviour of these cells under the conditions of the
experiment.*

From previous work done in the Heisenberg, Paluch and Raz lab, embryonic cells of
several zebrafish lineages show blebbing phenotypes both in the embryo and in culture
(Diz-Munoz et al., PLoS Biol. 8, e1000544, 2010; Maitre et al., Science 338, 253–6,
2012; Ruprecht et al., Cell 160, 673–685, 2015), so we think this is a normal phenotype.
We have added information about this in the fourth paragraph of the subsection
“Oscillations in isolated segmentation clock cells in vitro”.

We have included experiments investigating the expression of PSM genes. We examined
whether the cells in our cultures express No tail (Ntl, a zebrafish homolog of
Brachyury), a well-established marker of the posterior progenitor population (Martin and
Kimelman, Genes Dev. 24, 2778–83, 2010) and Tbx16 (Spadedtail), a marker of tailbud and
posterior PSM (Martin and Kimelman, Dev. Cell 22, 223– 32, 2012). Polyclonal antibodies
to these proteins had been previously published (Schulte-Merker et al., Development 116,
1021-1032 (1992); Amacher et al., Development 129, 3311-3323 (2002)), but these reagents
are limited or exhausted. Therefore, in the revised version of this paper we now
describe the generation and validation by our group of two new monoclonal antibodies to
these proteins in a new supplement (Figure 1—figure supplement 6).

During the original experiments that form the core of the manuscript, we had split the
starting material: one part was imaged as reported in the manuscript, the other part was
cultured in parallel and then fixed at later time points and immunostained for Ntl and
Tbx16 protein. We found that 81 out of 114 cells across 10 fields of view in the culture
had either elevated Ntl and/or Tbx16 staining in their nuclei after several hours of
culture. Since Ntl and Tbx16 expression is lost as cells move into the anterior PSM,
this finding argues that many of the cells in culture have maintained a relatively
posterior progenitor state. The dissection of the tailbud posterior to the end of the
notochord (see Figure 1 and Figure 1—figure supplement 1) is expected to recover cells of the
neural plate, skin, endoderm and lateral plate mesoderm, in addition to cells of the
PSM. Approximately 60% of the cells express Her1- YFP in culture (Figure 1), and we therefore anticipated that ~40% of cells in the
culture would not express markers of PSM. The finding that 71% of the cells express Ntl
or Tbx16 is consistent with most or all of the Her1-YFP positive cells expressing these
posterior markers. We have included information about the expression of Ntl and Tbx16
protein in the embryo and in vitro as a new supplement (Figure 1—figure supplement 7), and included text to summarize
these findings in the fourth paragraph of the subsection “Oscillations in isolated
segmentation clock cells in vitro”.

Summary information for each Fgf8b experiment in the current manuscript is provided in
table form ([Supplementary-material SD2-data]), and includes the number of expressing cells, cell survival, cell division,
etc. Cells that underwent division during the recordings had been previously included in
the touching category, because these cells tend to remain in close contact (see below:
reviewer 2, point 2), but in response to requests by reviewers 1 and 2, we have now
separated these cells, creating a category of dividing cells. The frequency of dividing
cells is low ~2%.

*3) What happened to the cells that touched each other? Do pairs of cells in
contact with one another synchronise their signalling, or oscillate with a shorter
period? Is there a community effect, and what is the minimum number of cells in a
group that is sufficient to maintain faster and synchronized oscillations?*

We appreciate these interesting questions, but this was not measured systematically in
our movies. The main problem was that the cells that touch each other usually crawled on
top of each other, making it difficult to distinguish between each individual cells
using wide-field imaging. We are currently exploring different culture, labeling and
imaging options to address this question directly and argue that such an analysis is
outside the scope of this study.

*4) Fgf8b has been added to the culture medium to mimic conditions in the
tailbud. What happens in absence of Fgf8b? Are there oscillations at all? Or damped
oscillations? or oscillations with even longer periods?*

The effect of Fgf8 was raised also by reviewer 2 (concern 1). Tailbud cells in our
culture oscillate without Fgf8b, also with a range of oscillatory phenotypes, but with
many fewer peaks on average. These peaks tend to occur at the beginning of the
recording, and in most cases appear to damp out. To document this, we have added to the
revised version traces from 52 cells from the same starting tailbud cell suspensions as
reported in the manuscript (experiments 240711 and 250112), but recorded in a separate
well of the divided dish without added Fgf8b. Peak counting analysis of these traces
found a median of 2 peaks, in agreement with our previously published work defining the
basic isolation and culture protocol (Webb et al., J Vis Exp (89) 2014), compared to a
median of 5 peaks for Fgf8b-treated cells ([Supplementary-material SD1-data]). Although period estimates for the
serum only cells are slightly noisier because of the fewer cycles, they are similar to
those of cells treated with Fgf8b. This phenotype is consistent with the proposed role
of FGF in the vertebrate tailbud as a factor that retains cells in a progenitor state
(Dubrulle, J. et al., (2001). Cell, 106(2), 219–232.; Diez del Corral, R. et al.,
(2002). Development 129, 1681– 1691). Rather than being instructive for the period of
oscillations, FGF appears to be permissive for the oscillatory state and we have now
included some discussion of this result in the fourth and sixth paragraphs of the
Discussion, as it is a unique test of various hypotheses about the role of FGF in the
embryo.

Thus, the addition of Fgf8b to this culture system keeps the cells oscillating
throughout the recording and allows us to observe more cycles per cell and improve
period estimates. We have included this information now in a new supplement (Figure 1—figure supplement 4), compare the
statistics of these cells directly in Figure 1—figure supplement 8, comment on the effects of FGF in the culture at the beginning of
the Results section.

*5) The legend to Figure 1—figure supplement 1 is not sufficient to understand the figure. There is both a red and a
grey line on each panel, but the legend confusingly only refers to a black line.
(What is the grey line? Background levels? This should be stated somewhere.) I
presume that the grouping of each set of traces refers to the independent replicates,
but this is not explained anywhere. The 'smoothed' traces should be labelled as such
on the figure. Also, the colour code is not correct* – *the smoothed
traces are in red with blue circles to indicate peaks and troughs, not in blue with
red triangles as indicated in the legend.*

We apologize for the mistakes in the color referencing. We have revised the figure
legend to accurately reflect the figure.

*Likewise, in Figure 1—figure supplement 2 and legend, the peak finding on the traces is actually labelled with red
triangles, not blue as indicated in the legend; filtering appears to be labelled with
blue triangles, not red as indicated. This lack of attention to detail has made it
difficult and time-consuming to review the manuscript.*

Again, apologies for the mistakes in the color referencing. We have revised the figure
legend to accurately reflect the figure.

*6) Figure 1—figure supplement 6* –

*these traces show timecourses for 10 individual isolated cells. Have these data
been subjected to the same smoothing, peak and trough analysis etc. as the data from
the low density set? Do they behave in a similar way?*

We have updated this figure (now Figure 1—figure supplement 9) to show the raw, background-subtracted data with smoothing and
with peak calling using the same basic analysis pipeline as the other time series. This
is described in the figure legend. The statistics from these cells are now included in
tabular form (Figure 1—source data) for
comparison with cells in all conditions. Although the certainty associated with these
values is lower because of the smaller sample size, the cells behave in a similar
way.

*Reviewer #2: The manuscript presents a very worthwhile data set, detailing
time-course expression profiles for a Her1 reporter with high temporal resolution.
These show a high degree of variability, and a range of analyses are performed to
extract phase and amplitude data. The authors go on to compare these summary measures
to those obtained from a generic model of a system close to a supercritical Hopf
bifurcation (at the boundary between decaying and sustained oscillations). They show
that reasonable (though not perfect) agreement between the two sets of measures
(experimental and simulated) can be achieved for an appropriate set of model
parameters. These results are interesting in their own right, as a careful
investigation of the nature of the time-courses obtained from the cultured tailbud
cells. I'm less convinced that they say anything particularly important about
somitogenesis in zebrafish. My primary concern here is that the Her1 expression
observed in the low-density cultured cells is so distinct to that observed in tailbud
cells in an intact embryo. The period of oscillation in vivo is 27min; the mean
period extracted from the time-courses of the cultured cells is around three times as
long as this (ca. 78min). The two sets of oscillations thus represent markedly
different dynamics. The long-period noisy oscillations observed in cultured cells may
(or may not) be due to a dynamical system close to a Hopf bifurcation, but if they
are, it is hard to see how this relates to a dynamical system based on the same
components oscillating in a tissue with almost three times the frequency! The key
observation in this regard is that the somitogenesis period in explanted PSM is
around 55min, rather than the 27min in the embryo at the same temperature. This is
reported in this manuscript without any real discussion. But surely this is a
striking and potentially important result. My overall assessment is that the data set
and data analysis are high quality and worthy of publication. The potential fit to a
very generic model is also interesting, but I'm not convinced that the fit is
actually that good. Given the large discrepancy between the cultures and the embryo,
I'm not convinced that the data have much to say about somitogenesis.*

We share with the reviewer the view that the slowing of segmentation in the explant is a
striking and potentially important result. Of course, any in vitro model of an in vivo
situation must be viewed with caution, and zebrafish primary cell culture is in its
infancy, particularly when it comes to understanding dynamics. However, we disagree that
the difference in period between intact embryo and individual cell is in itself a
substantive problem with the in vitro model; we argue that the differences may allow us
to begin to reveal and estimate the magnitude of key interactions present in the tissue
that are missing in vitro. There are a number of points raised by the reviewer in the
paragraphs above, which we attempt to clarify and discuss here, as well as describe the
new experiments and revisions we made.

1) Comparison of period lengths. Given the previously noted slowing of development in
explanted zebrafish tissue (Langenberg et al., Dev. Dyn. 228, 464–474 (2003)), we needed
a more direct measure of this slowing to get a better comparison of the differences
between the genetic oscillations in the tissue and single cell. We have therefore now
added a new experiment in which we estimate period directly from the YFP signal in
explanted Looping1 tailbuds (Figure 1—figure supplement 3, [Supplementary-material SD1-data]). We found a period of 42 minutes in the explant, which is about
1.5 times the longer than the period of segmentation in the intact embryo. When cells
are dispersed from the tailbud explants in culture, the period lengthens approximately
2-fold (regardless of the presence of FGF). We argue this is a direct measurement of the
difference between tissue-level and single cell oscillations in culture. An increase in
period after dissociation was expected but its magnitude was not, so these new
experiments provoked by the reviewers’ critiques may have revealed a new phenomenon. We
speculate that this difference might be due to an as yet unknown additional source of
coupling in the tissue, and/or a tonically-expressed factor in the tailbud that elevates
the frequency in the tissue, but is diluted once the cells are dispersed. Please also
see the detailed response to reviewer 1, substantive concern 1, above. Thus, the revised
manuscript removes the lengthy and somewhat abstract comparisons of estimates in the
Results, replacing them with measurements at the beginning of the Results, and a
discussion of the differences at the end.

2) With respect to the more general point about whether a system can oscillate over a
range of 2-3 fold in frequency using the same components, we would point out that
changes in temperature cause the whole system to shift its frequency 3-fold, while
generating a normally proportioned embryo (Schröter et al., Dev. Dyn. 237, 545–553,
2008). Such reliability at different temperatures and periods is likely advantageous for
an embryo that grows up without parental care and is subject to a fluctuating
environment. Although this hasn’t been examined in detail, we would be hesitant to say
that the oscillations at one of the temperatures do not relate to the dynamics of the
system at other temperatures. Whatever the changes to the biochemistry, these end up in
the parameter omega for the frequency in the model. The model is used primarily to probe
the noise in the data and doesn’t describe the changes in time-scale. Thus, we argue
that the change in frequency is not in itself a problem for the model or
interpretations.

We don’t yet understand the 1.5-fold slowing in the explants relative to the intact
embryo. Our hypothesis is that some general temperature-like factor is influencing the
rate of development generally. It was reported that changes in O2 can slow zebrafish
development and even temporarily arrest it (Padilla and Roth, 2001, PNAS 98(13):7331-5),
so gas exchange may play a role.

Alternatively, there may be some rate influencing metabolic factor that is normally
available from the yolk that is missing in the explants, causing the tissue to use an
alternate energy source with a lower flux, which influences the general developmental
rate. We wish to investigate this slowing in the future, as it is indeed striking and
could well be more generally important, but we have argued here that it should not
present an insurmountable challenge to the current analysis and interpretation of the
data.

We are confused by the reviewer’s statements “…reasonable (though not perfect) agreement
between the two sets of measures (experimental and simulated)” and “…I'm not convinced
that the fit is actually that good”. It is rare that model and experimental data are in
perfect agreement, and we are deliberately using a very simple model to try to tease
apart the basic timescales in the noise. It would almost certainly be possible to add
more complexity to the model (additional parameters), or change its structure
altogether, and thereby improve the match between model and experiment (Note that in
Figure 2, there is no fitting in the technical
sense), but it is unclear what would be gained. Nevertheless, it is important to point
out what the model did not do a good job of explaining, and in this respect we have
revised the paragraph comparing model and data (see point 3 below).

*Substantive concerns: 1) Fgf8b was added to the culture medium "to mimic
the signaling environment of the tailbud". Was this necessary to see
oscillations in Her1-YFP levels? What happens if it is not added?*

See response to reviewer 1, specific comments 4, above.

*2) Did cells in culture never divide? This is not mentioned (other reasons for
discounting time-courses are mentioned). Cells in the tailbud typically undergo a
single division at around 15-16hpf (see Figure 1 in Bouldin et al. (2014). Genes & Development, 28, 384-395.) It
seems that this division should fall within the time window of the current study. Are
the culture conditions blocking this division, and if so, might that be expected to
affect the oscillations?*

Cells divide during the recording, although this is at low frequency. This behavior was
previously included in the “touch another cell” category, as after division the cells
almost always stay in contact and crawl over each other, but [Supplementary-material SD2-data] has
now been revised to include the division category explicitly.

The number of cell divisions we see is reduced relative to what was measured in vivo by
Bouldin et al., 2014, and therefore it seems that the culture conditions are inhibiting
division. Importantly, Zhang et al., 2008 (Cell cycle progression is required for
zebrafish somite morphogenesis but not segmentation clock function. Development,
135(12), 2065–2070) examined the emi1 mutant zebrafish that lacks cell divisions after
early gastrulation and concluded that the segmentation clock was normal. Bouldin showed
that a heatshock of cdc25a expression increased the mitotic rate and prolonged Tbx16
expression, but did not present evidence about the effects of reducing division on
differentiation. On this basis, we wouldn’t have expected that reducing cell division
would affect the cells ability to oscillate per se. In any case, we have checked Tbx16
and Ntl expression in the cells of parallel cultures (as above) and found that
expression is consistent with the tailbud. We argue therefore, that although the culture
system appears to reduce cell division, this has not affected the differentiation state
of the cells.

From previous studies in the embryo (Delaune, E. A., et al., (2012).
Single-cell-resolution imaging of the impact of Notch signaling and mitosis on
segmentation clock dynamics. Developmental Cell, 23(5), 995– 1005), divisions were
observed to introduce phase noise into the time series. Thus, our analysis, which
excludes cells that divide, is likely to give a slightly higher measurement of precision
than if we included dividing cells. Additionally, because of the very low rate of cell
division in the culture, we argue that cell division cycle stage is unlikely to be a
major contributor to the heterogeneity we observe. We have commented on these aspects in
the first paragraph of the subsection “Precision of persistently oscillating cells”.

*3) The authors state (subsection “Heterogeneity in the population of oscillating
cells”) that the simulated data are "in good agreement with the experimental
data (Figure 2)" However, Figure 2 don't match very well, as noted
by the authors themselves in the next sentence. There is clearly more overall
variability in amplitude in the real data (2B) than in the simulated data (2F),
though the correlation of neighbouring peak amplitudes is similar. So the data show
similar short-time coherence of amplitude to the model, but significantly greater
variability in amplitude overall.*

As mentioned above, we have deliberately used a very simple model, where we did not
expect an exact match to the data. In order to avoid giving the impression that we have
captured all the features of the data, we have replaced the phrase “in good agreement”
with “comparable to”, and then given a sentence where we point out the differences to
the data:

“Slow amplitude fluctuations can drive the oscillators in and out of the oscillatory
state (Figure 2; Figure 2—figure supplement 2), and introduce correlations in the
amplitude of consecutive cycles that are comparable to the experimental data (Figure 2). Interestingly, the trend to higher
amplitude variance at higher amplitude values, and the existence of a low occurrence of
high relative amplitude cycles are not captured by the theory.”

*4) The data do appear to support the idea that the main variability in the
oscillations is in the amplitude rather than in the period (Figure 2). The authors use this finding to focus their
attention on a S-L model with q=0 (so, the frequency is independent of the
amplitude). This restricts the type of supercritical Hopf bifurcation that could
underlie the observed oscillations (the normal form does not require q=0). Does this
restrict the possible molecular mechanisms underlying the observed Her1 oscillations?
In particular, do noisy negative feedback oscillators behave in this way? It is my
understanding that stochastic simulations of negative feedback circuits (like the
ones alluded to by the authors) that exhibit sub-Hopf stochastic resonance do show
amplitude-dependence of the period (increased amplitude leads to increased period).
Furthermore, the authors have previously shown that the use of the Her1-Venus
transgene increases the period of oscillations. Is this likely to be due to the
overall increase in expression level of Her1 in the Looping line? The
amplitude-dependence of period (or lack of it) is potentially important for the
tissue-wide oscillations observed in the PSM, as the value and distribution of q is
an important factor in the ability of oscillators to synchronise.*

Our choice of q = 0 was made partly for simplicity, and also based on some data that we
didn’t present before, which is the correlation between period and amplitude – apologies
for this. We have now included this plot in Figure 2—figure supplement 1 as panel D. The reviewer is correct to point out that in
general when a system is reduced to its normal form close to a Hopf bifurcation, it
requires a special non-generic symmetry to get one of the parameters = 0. Thus, we do
not make any claim that the underlying system has this property; in general we expect
that q would not be zero. However, with noise in the system, a small value of q can
easily be blurred. We show this scenario in Figure 2—figure supplement 1 panels E-I, in which we examine the effect of noise on
the SL model with known values of q.

We also examined the behavior of the classical Lewis negative feedback model (Lewis, J.
(2003) Autoinhibition with transcriptional delay: a simple mechanism for the zebrafish
somitogenesis oscillator. Current Biology 13(16), 1398–1408). For this model also, using
the parameters given in that paper noise can blur the correlation between the amplitude
and period. We agree that using these new data to constrain and develop microscopic
models of the segmentation clock, thereby probing the molecular mechanisms at the core
of the clock is a very interesting prospect, but to do so will require a systematic
examination of how different sources of noise affect the dynamics of the various models,
as well as examination of various perturbations. We argue that this is a major new
project, and outside the scope of the current paper.

In the interests of furthering the utility of this data, we have now included a
spreadsheet in which the raw and background-subtracted data from all low-density
oscillating cells is presented so that other researchers may analyze this in detail and
test against other theoretical descriptions ([Supplementary-material SD3-data]).

The overall increase of period in the Looping line is unlikely to be due to the elevated
levels of Her1 in the embryo. We have generated several independent lines with the same
transgene and these express a range of levels, but have an identical period offset to
wildtype (10 ± 2%) as the published Looping line.

Reviewer #3: Webb et al. describe segmentation clock oscillations of individual
dissociated zebrafish tail bud cells. They find that oscillations are present in
vitro (as expected), but that the period is longer than in vivo (as previously
postulated but shown here for the first time). A noise model can account for the
degeneration of oscillations. The data is interesting for zebrafish researchers but I
am not convinced that it goes beyond previous studies in chick and mouse. It seems
straightforward to compare the in vitro oscillations to the in vivo oscillations in
Notch signaling mutants. This experiment would test if the differences in vitro can
be explained by the absence of Notch signaling.

As described above, this study clearly goes well beyond previous work in mouse and
chick.

We are not aware of any report of reliable imaging and tracking of oscillations in
tailbud cells, the subject of this manuscript, in an intact embryo. Delaune et al. (Dev.
Cell 23, 995–1005, 2012) and Shih et al. (Development 142, 1785–1793, 2015) have
reported oscillations of single cells in the bulk of the PSM, where cell movement in the
tissue is very limited, enabling reliable semi-manual tracking over time. These cells
are all systematically slowing their oscillations (Morelli et al., HFSP J. 3, 55–66,
2009; Shih et al., 2015) and are not the relevant comparison to our cell culture. In
contrast, cells are moving rapidly in the tailbud (in x, y and z) (Lawton et al.
Development, 140(3), 573–582, 2013). We have not yet been able (nor have other
laboratories) to extract extended tracks on the timescale of several cycles that would
enable a reliable estimate of the period of tailbud cells in the intact embryo. This is
why we have considered the period of oscillations at the local tissue level instead
(effectively averaged over many cells moving within the tailbud) where the measurements
are reliable (Soroldoni et al., Science 345, 222–225, 2014). Note that this period would
include the effects of local and tissue-level collective processes such as cell-cell
coupling or diffusive signaling.

[Editors’ note: the author responses to the re-review follow.]

*Reviewer #1: Firstly, and also with respect, I would like to clarify that I did
not state that the previous manuscript was 'only a technical' advance over previous
work; in fact, this part of my review was a positive comment in support of the paper.
The authors have addressed all the points that I raised, most of them very
satisfactorily. I think the argument is clearer and that the additional data (e.g.
maintenance of marker expression) help with the overall interpretation. My one
remaining concern relates to the issue of dividing cells in the low-density dataset.
It is helpful that the authors have given more detail on cell divisions in a source
file, and I appreciate the argument that the focus of the study is necessarily on the
single cells. However, more explicit information is still needed here, especially for
Video 1. The legend to Video 1 indicates that it corresponds to the
field of view for the non-dividing cell in Figure 1. It would help to mark the cell of interest (e.g. with a ring or
arrowhead) at the start of the video. Otherwise, the text in the fifth paragraph of
the subsection “Oscillations in isolated segmentation clock cells in vitor”, leads
the reader to think that Video 1 is
illustrating a whole field of cells that do not divide.*

We have marked the cell of interest with an arrow at the beginning of Video 1, and we have re-written the legend to
Video 1 to describe what is seen and how
we processed the different cellular behaviors for analysis in the paper. We hope that
this description, along with other changes (see below), will help to clarify the
reported cell categories and to reduce ambiguity about the decision-making process.

*More importantly, however, in addition to the non-dividing cell in the lower
left, Video 1 also includes no fewer than
five clear examples of cell divisions, four of them YFP+ (at 00.26 (top right and
lower right), 00.30 (middle left), 00.33 (extreme top right) and 00.41 (bottom
right)). I find this very surprising, given the source data file ([Supplementary-material SD2-data]),
which documents only 9 instances of cell division in the entire dataset from a total
of 29 fields of view, which would suggest that cell division events are rare. The
observation of five division events in a single field of view also suggests that
Video 1 is not representative of the
dataset as a whole, and is directly contradictory to the statements in the rebuttal
that 'it seems that culture conditions are inhibiting division'; 'because of the very
low rate of cell division in the culture' etc. In short, I am not confident that the
authors have measured the rate of cell division accurately in their cultures. Further
explanation is required here.*

We re-counted all the divisions in the movies, and added an expanded explanation of how
cells were categorized. As we now state in the legend to [Supplementary-material SD2-data]:

“Across the 29 fields recorded, we observed cell divisions in both YFP-negative (30, 5%
of total cells) and YFP-positive cells (13, 2% of total cells). We found a range in the
number of cell divisions from 0 to 5 cells per field, with an average of 1.5 ( ± 1 SD)
divisions per field. The categories of disqualification list the *first*
event in a recording that led to disqualification. For example, four divisions in
YFP-positive cells occurred after the cell had been disqualified for another reason
(movement in and out of field, touching another cell).”

In summary: (1) we re-counted the same number of YFP-positive divisions as we had
previously reported (2%); (2) a slightly larger number of YFP-negative cells are
dividing throughout the experiments (5%); and (3) the field with the highest number of
divisions (5) was indeed the field shown in the supplementary video.

This means that Video 1 is not representative
of the mean number of dividing cells per field in the data set as a whole, but if one
calculates the approximate Maximum Load expectation for throwing 43 balls (cell
division) into 29 pots (field of view), it’s ~4.5, so we don’t think a field with 5
events was unexpected. We argue to retain this video for illustrative purposes, with
appropriate annotation, as it does give a nice, compact overview of the different
behaviors that are seen in the experiment, as highlighted by Reviewer 1 (below). We hope
that readers will find these interesting and it will stimulate future studies in the
community.

We have now revised the Results section, Figure 1 legend, and the legend of the movie to reflect these facts, and in
particular to avoid giving the impression that the field is representative of the mean
division rate per field in the entire data set. Results subsection “Oscillations in
isolated segmentation clock cells in vitro”, fifth paragraph, Figure 1 legend, Video 1 legend.

*As the authors have pointed out in their rebuttal (although I could not find
this in the manuscript), the pairs of daughter cells remain attached to one another.
As far as I can see, there does not appear to be a consistent timing of division
relative to the phase of the oscillation between the three examples, or maintenance
of synchrony between daughters within a pair, although oscillation in at least one
daughter can clearly continue after division. For example, after the division at
00.30, the daughter on the left of the pair has a very clear YFP peak at 00.39-00.40,
while its sister on the right has very low YFP signal at the same time point. Since
the dataset captures this very interesting information, it deserves some comment,
even if a full analysis is beyond the scope of the current study.*

We have commented on the cells’ tendency to remain attached after division and the
interesting observation that oscillations can continue after division in the Results
section, subsection “Oscillations in isolated segmentation clock cells in vitro”, fifth
paragraph.

*For cells that did divide (including the four YFP+ cells shown in Video 1), were oscillations measured up to
the point of division, or were these cells eliminated from the analysis altogether?
The text is not altogether clear on this point.*

Any cell that divided at any point in the recording was disqualified from analysis. We
now state this explicitly in the fifth paragraph of the subsection “Oscillations in
isolated segmentation clock cells in vitro”.

*Reviewer #2: I continue to believe that the data described in this manuscript
are interesting and valuable, and I think it is very positive that the authors are
making the raw data available to allow others to analyse them (although I maintain
that the authors have done a good job of this already). The authors have addressed my
queries and concerns. The additional data, and provision of raw data, make it easier
to assess the validity of the statements made in the manuscript. I have one remaining
niggle with the conclusions drawn from the data and the corresponding nomenclature
employed by the authors. The cells are derived from the tailbud and maintain tailbud
markers (new data). The cells certainly exhibit oscillations, which have features
that are consistent with an underlying dynamical system that is close to a Hopf
bifurcation. So it is fair to say that the cells are oscillators. However, I remain
to be convinced that these facts are sufficient to call the cells "single
zebrafish segmentation clock cells" (e.g. sixth paragraph of the Introduction
and many other places). The segmentation clock is surely a coordinated tissue-wide
oscillatory operating in the tailbud/PSM. A parsimonious (though not necessary)
explanation of the findings reported is that the segmentation clock results from the
interaction of some noisy cell-autonomous genetic oscillators. But that is not
sufficient to call the oscillations in the single cells in vitro a "segmentation
clock". I accept the argument that the period of the in vitro segmentation clock
can exhibit marked temperature-dependence. But that does not lessen the difficulty of
arguing that if a cell-autonomous oscillation in tailbud-derived cells is
borderline-Hopf with a period of ca. 80min, then the in vivo clock, with a period of
ca. 27min, has something to do with borderline-Hopf dynamics. I maintain that I do
not think it is easy to explain how an oscillatory mechanism can be close to a Hopf
bifurcation at two radically different periods. It may be that the mechanisms
underlying the observed in vitro cell-autonomous oscillations and the in vivo
segmentation clock are closely related (and it is probably true), but that does not
mean that their dynamics are similar. Just because cells can oscillate in two
different conditions does not mean that one can say that the oscillations are two
different versions of the same thing. In summary, I like this study. I think the in
vitro oscillations are interesting and the analysis is powerful. I just don't like
the idea of calling these oscillations a segmentation clock* –

*that is what operates in the embryo.*

We fully agree with the reviewer’s assessment. Indeed, our group has often argued for
the use of the term segmentation clock to refer to the tissue-level rhythmic patterning
system in the embryo, so we should have been aware of this connotation. We have changed
the term “segmentation clock cell” to “cell isolated from the segmentation clock” or
equivalent and appropriate wording throughout the paper (once in the Abstract, three
times in the Introduction, eight times in the Results, and once in the Discussion (see
annotated manuscript version). We have also inserted a qualifying sentence about the
general slowing in the Discussion:

“Nevertheless, until the mechanism of this general slowing in vitro and its influence on
the molecular and cellular processes within the segmentation clock are understood, we
must remain circumspect in our interpretations.”

Reviewer #3: Webb et al. have revised their manuscript to more clearly explain
the limitations of previous studies in concluding that cell-autonomous oscillations
form the basis of the segmentation clock. The differences between single cells,
tissue explants and in vivo tail buds are striking but unexplained. The paper is a
nice technical advance but the main conclusions were expected from previous studies,
and the most interesting phenomena of different periods in vitro and in vivo remains
unaddressed.

We agree that the differences between single cell and tissue level activities are still
not understood. We also agree that some theoretical studies gave expectations that are
now supported by our work (and other theories are now disqualified), but we continue to
argue that there was no previous experimental work of this kind, and that the
experimental demonstration and characterization of the autonomous state is an important
contribution.